



# Simulating interactive ice sheets in the multi-resolution AWI-ESM 1.2: A case study using SCOPE 1.0

Paul Gierz[1], Lars Ackermann[1], Christian B. Rodehacke[1, 2], Uta Krebs-Kanzow[1], Christian Stepanek[1], Dirk Barbi[1], and Gerrit Lohmann[1, 3]

[1] Alfred-Wegener-Institut, Helmholtz-Zentrum für Polar- und Meeresforschung, Bremerhaven, Germany
[2] Danish Meteorological Institute, Copenhagen, Denmark
[3] University of Bremen, Bremen, Germany

**Correspondence:** Paul Gierz (pgierz@awi.de)

**Abstract.** Interactions between the climate and the cryosphere have the potential to induce strong non-linear transitions in the Earth's climate. These interactions influence both the atmospheric circulation, by changing the ice sheet's geometry, as well as the oceanic circulation, by modification of the water mass properties. Furthermore, the waxing and waning of large continental ice sheets influences the global albedo, altering the energy balance of the Earth System and inducing climate-ice

sheet feedbacks on a global scale as evident in Pleistocene glacial-interglacial cycles. To date, few fully comprehensive models exist, that do not only contain a coupled atmosphere/land/ocean component, but also consider interactive cryosphere physics. Yet, on glacial-interglacial and tectonic time scales, as well as in the Anthropocene, ice sheets are not in equilibrium with the climate, and prescribed fixed ice sheet representations in the model can principally be only an approximation to reality. Only climate models, that contain interactive ice sheets, can produce simulations of the Earth's climate which include all feedbacks

and processes related to atmosphere-land-ocean-ice interactions. Previous fully coupled models were limited either by low spatial resolution or an incomplete representation of ice sheet processes, such as iceberg calving, surface ablation processes, and ocean/ice-shelf interactions. Here, we present the newly developed AWI-Earth System Model (AWI-ESM), which tackles some of these problems. Our modelling toolbox is based on the AWI-climate model, including atmosphere and vegetation components suitable for paleoclimate studies, a multi-resolution global ocean component which can be refined to simulate

regions of interest at high resolution, and an ice sheet component suitable for simulating both ice sheet and ice shelf dynamics and thermodynamics. We describe the currently implemented coupling between these components, present first results for the Mid-Holocene and Last Interglacial, and introduce further ideas for scientific applications for both future and past climate states with a focus on the Northern Hemisphere. Finally, we provide an outlook on the potential of such fully coupled Earth System models in improving representation of climate-ice sheet feedbacks in future paleoclimate studies with this model.





# 1 Introduction

With the growing impact of anthropogenic activity on the Earth's climate, it has become increasingly important for climate models to be able to provide comprehensive representations of the Earth system in order to project the climate changes which may occur in the coming decades and centuries, and to enable understanding of the involved mechanisms, feedbacks, and

responses. An integral part of Earth's climate is the cryosphere, which contains vast amounts of the planet's freshwater supply (sea-level-equivalent of Antarctica, Greenland and mountain glaciers are $58.3\,\mathrm{m}$, $7.36\,\mathrm{m}$, and $0.41\,\mathrm{m}$, respectively, Vaughan et al., 2013). Recent observations have suggested that the Greenland Ice Sheet (GrIS) has seen an acceleration of surface melt and mass loss in recent years (Hanna et al., 2019; Khan et al., 2014; Shepherd et al., 2012; Smith et al., 2020).

Aside from the impact of ice sheet melt on both local and global sea-level, interactions between the continental ice sheets and

the remainder of the Earth system have the potential to induce large scale reorganization of both the large-scale atmospheric (Löfverström and Lora, 2017; Roberts et al., 2019) and interior ocean circulation (Gong et al., 2015). Under future warming scenarios, this leads to a slowdown of the Atlantic Meridional Overturning Circulation (AMOC) and a consequent relative retention of heat in the tropical latitudes, thus relatively cooling the North Atlantic (Fichefet, 2003; Gierz et al., 2015; Rahmstorf et al., 2015). Mass loss from the continental ice sheets has also played a critical role in Earth's past climate, in particular during

the last $800{,}000\,\mathrm{years}$, where paleoclimate proxies provide evidence for repeated, abrupt temperature shifts (Barker et al., 2011, 2015). These shifts are likely linked to corresponding changes in the continental ice sheets. To which extent mass loss from the ice sheets is a trigger or a response is still under debate within the scientific community (Barker et al., 2015).

In this paper, we address the need for Earth System models with a capability of simulating the interactions involved in such processes in a comprehensive manner. We present the AWI Earth System Model (AWI-ESM) in an updated version that is based on AWI-CM (Sidorenko et al., 2015) and AWI-ESM-1-1 (Shi et al., 0; Lohmann et al., 2020, Under Revision).

The AWI-ESM-1-2 includes an interactive ice sheet/ice shelf component. We describe the coupling between the ice sheet model and the remainder of the climate system which are implemented in our model. Furthermore, in order to evaluate the performance and characteristics of the coupled AWI-ESM-1-2 climate – ice sheet system, we show case studies for several warm climates states simulated with this model, specifically with reconstructed or projected sea level that is higher than at present. The simulations based on the CMIP6 core experiments (DECK simulations) (Eyring et al., 2016) which cover the

Pre-Industrial equilibrium control simulation, a transient scenario for $1\%\,/\,\mathrm{yr}$ increase in atmospheric carbon dioxide, and a simulation with abrupt increase of carbon dioxide concentration to four times the Pre-Industrial level. Furthermore, we conduct two of the Paleoclimate Model Intercomparion Project (PMIP4) time slice simulations described by Otto-Bliesner et al. (2017), those for the Mid Holocene (MH, $6{,}000\,\mathrm{years}$ before present) and for the Last Interglacial (LIG, $127{,}000\,\mathrm{years}$ before present,

henceforth $\mathrm{ka\,BP}$). The PMIP4 interglacial climate states are excellent test cases for how our model performs in warmer-than-present situations, where ice sheet's runoff has the potential to influence the surrounding ocean dynamics: both the LIG and MH are characterized in the PMIP4 model ensemble by pronounced warmer then Pre-Industrial boreal temperatures in summer, and by a slightly warmer annual mean (Brierley et al., 2020a; Otto-Bliesner et al., 2020). Hence, they enable the study of GrIS dynamics and the related impact on climate in a comparably warm setting that is similar (but not a perfect analogue) to climate





conditions expected in the near future (Otto-Bliesner et al., 2017). Based on our results, we discuss insights and limitations which must be considered when using a coupled climate/cyrosphere model, in particular with regards to model initialization and spin-up. We conclude with an outlook for further studies of glacial climate states. A companion paper (Ackermann et al., under review), examines the influence of interactive ice sheets on the AMOC under the AR4 IPCC scenarios.

In the next section we describe the primary building blocks of AWI-ESM-1-2. Thereafter, we demonstrate the coupling
interface between the climate and the ice sheet models, showing the exchanged fields. Next, we perform several climate experiments using our new model, and demonstrate its performance for the PMIP scenarios. Lastly, we provide an outlook of new scientific questions which could now be addressed using AWI-ESM-1-2.

## 2 Methods and Model Description

In this section we describe the details of our newly developed modelling system, namely the AWI-ESM-1-2 coupled climate–
vegetation – ice sheet model. The AWI-ESM-1-2 is based on two numerical models that are established tools: the state-of-the-art comprehensive Earth System Model AWI-ESM-1-1, that itself is the paleoclimate version with dynamic vegetation of the AWI-CM-1-1 that has been described by Sidorenko et al. (2015), and the Parallel Ice Sheet Model (PISM, version 1.1) described by Bueler and Brown (2009). The AWI-ESM-1-1 is a CMIP6/PMIP4 model and has been employed in PMIP4 for multimodel-intercomparison of the Mid-Holocene (Brown et al., 2020; Brierley et al., 2020a), the Last Glacial Maximum
(Brown et al., 2020; Kageyama et al., 2020), and the LIG (Brown et al., 2020; Otto-Bliesner et al., 2020). Towards facilitating coupling of climate and cryosphere model components, the coupling interface — SCOPE, the **S**tandalone **CO**u**P**l**E**r — has been developed. This novel tool is described for the first time in our publication. Below we give an overview on characteristics of the model components that comprise the AWI-ESM-1-2.

### 2.1 The Earth System Model AWI-ESM-1-1: Climate Components ECHAM6, JSBACH, and FESOM

The AWI ESM without coupled ice sheets, hereafter AWI-ESM-1-1, is the climate component in our comprehensive climate/cryosphere system. It consists of the sixth generation of the European Centre for Medium-Range Weather Forecasts' Model in Hamburg (ECHAM6) atmospheric general circulation model (Stevens et al., 2013), the Jena Scheme for Biosphere-Atmosphere coupling in Hamburg (JSBACH) land surface and carbon cycle model with dynamic vegetation (Brovkin et al., 2013; Reick et al., 2013), which is an integral part of the ECHAM6, and the Finite-Element Sea ice-Ocean Model (FESOM)
(Danilov et al., 2004; Timmermann et al., 2012; Wang et al., 2014). The atmosphere/land-biosphere and ocean components are coupled via the OASIS3mct library (Valcke, 2013; Craig et al., 2017). The coupled atmosphere/land – ocean model setup is described in detail by Sidorenko et al. (2015), with the difference that our model version employs vegetation dynamics. This section briefly describes each of the components of the AWI-ESM-1-1.

The ECHAM6 atmosphere model is an adaptation of the European Centre for Medium-Range Weather Forecasts' (ECMWF)
weather model. We use ECHAM6 version 6.3.04p1 for our setup. It is based upon a spectral dynamic core, and simulates the atmosphere from the Earth's surface to the top of the model atmosphere based on a hybrid sigma-pressure vertical coordinate.





The bulk of the vertical discretization resolves the troposphere. Yet, with the uppermost layer at a pressure level of $0.01\,\mathrm{hPa}$, which is roughly up to an altitude of $80\,\mathrm{km}$ above the Earth's surface, this high-top model resolves beyond the troposphere also the stratosphere, and partly covers the mesosphere within its 47 layers. The version employed in this work uses a horizontal

triangular wave truncation of T63 in the spectral core, corresponding to a lateral resolution of approximately $1.8° \times 1.8°$, equivalent to a grid cell length of $210\,\mathrm{km}$ at the equator and about $100\,\mathrm{km}$ zonally in Greenland.

Incorporated into the ECHAM6 model is JSBACH v. 3.11, which computes land surface processes, including the carbon cycle, vegetation dynamics, and a lateral routing scheme for continental runoff towards the coast that is based on the work by Hagemann and Gates (2003). JSBACH uses the same lateral resolution as the overlying atmospheric model. It is inherently

coupled to the atmosphere component and any coupling of fluxes between atmosphere and land surface is performed inside ECHAM6. From a perspective of the simulation of ice sheet dynamics and of the relevant climate feedbacks, JSBACH has an important task in resolving physical processes that change the energy balance of the land surface. JSBACH is the component of the ECHAM6 model that is responsible for albedo calculations. It is able to differentiate between albedos for both visible light as well as near-infrared radiation. Furthermore, it is able to differentiate between various types of bare soil and vegetation

classes, of which a total of 13 are distinguished in our model setup. Each of these vegetation and bare soil types provide distinct properties for surface albedo and water carrying capacity, thus influencing both the energy and mass balance of the simulated region. Our simulations use 8 of these types, which are dynamic and can geographically shift their coverage based upon the prevailing climatic conditions. These 8 types are summarized in Table 1.

The FESOM is a finite element ocean and sea ice model (Danilov et al., 2004; Wang et al., 2014). A strength of the finite

element approach is that it enables grid refinement, and hence allows to locally increase mesh resolution while limiting the computational expense in the global domain. As a result, computational power can be focused on critical regions of the ocean, where small-scale dynamics occur that impact on the large-scale circulation. In the setup employed for this study, we choose to increase the resolution across the entire North Atlantic, the coastlines, the equator, and the Southern Ocean, ensuring good representation of several key oceanographic processes — including deep and bottom water formation, equatorial dynamics, as

well as coastal boundary currents. Figure 1 shows the grid, respectively mesh, resolution used for both ECHAM6/JSBACH and FESOM. This resolution is refered to as "low resolution", and abbreviated as LR, thus yielding the complete model name AWI-ESM-1-1-LR.

Internal coupling between ECHAM6/JSBACH and FESOM is performed via the OASIS3-mct coupler (Valcke, 2013; Craig et al., 2017). Exchanges of mass and momentum between the oceanic and atmospheric components occurs every $6\,\mathrm{hours}$. Our

model setup builds upon the work by Sidorenko et al. (2015), who present the coupled model setup ECHAM6/JSBACH/FESOM, albeit without dynamic vegetation.

## 2.2 The PISM Ice Sheet Model

The Parallel Ice Sheet Model (PISM) uses a hybrid system (Bernales et al., 2017) that combines the Shallow Ice Approximation (SIA) and Shallow Shelf Approximation (SSA) to solve the equation of motion (Greve and Blatter, 2009). The Shallow Ice

Approximation reproduces the ice flow of grounded ice frozen to the bedrock. The Shallow Shelf Approximation describes


floating ice shelves and acts as an effective "sliding law" for ice streams (Bueler and Brown, 2009). The latter exploits the yield stress given by a Mohr-Coulomb formulation that mimics the basal resistance of plastic till (Schoof, 2006). The model considers glacial isostatic adjustment (GIA) of a viscoelastic deformable Earth (Bueler et al., 2007; Lingle and Clark, 1985) and takes into account the geothermal heat flux driven by the distribution of Shapiro and Ritzwoller (2004).

The Surface Mass Balance (SMB), which determines the accumulation and ablation (which is ice mass loss) of ice at ice sheet surface, is implemented by the Positive Degree Day (PDD) method within PISM, which is a temperature index scheme based on (Calov and Greve, 2005; Reeh, 1991). In this scheme, the total number of days multiply by the excess above the freezing temperature ($273.15 \, \mathrm{K}$) are integrated. Degree-day-factors (DDFs), appropriate for snow ($5 \, \mathrm{m \, K^{-1} \, day}$) and ice ($8 \, \mathrm{m \, K^{-1} \, day}$), are applied in order to estimate the amount of surface ablation. As the climate model only saves

monthly averaged climate characteristics, forcing for the ice sheet model adds white-noise in order to generate synthetic daily temperatures as an ice sheet forcing.

    In order to facilitate simulation of sea-proximal ice sheet regions and to realize the coupling between ocean and floating ice shelves, the setup additionally includes three calving parameterizations. First, thickness calving occurs at marginal ice shelf grid points with a thickness of less than $200 \, \mathrm{m}$. Second, continental-shelf calving is applied to ice that crosses the $2000 \, \mathrm{m}$-

depth bedrock contour line (contemporary bedrock elevation of ETOPO1, Amante and Eakins, 2009). Finally, we employ the Eigencalving parameterization, which uses stress field divergence with a proportionality constant of $1 \cdot 10^{17}$, contributes to the calving. The grounding line position (that is the transition between the grounded ice sheet and floating ice shelves) is handled by a sub-grid scheme described by Feldmann and Levermann (2015). For the basal conditions of floating ice shelves, various parameterizations are available, which are described below in the section about the coupling between the ocean and ice

sheet/ice shelf. Specifically, AWI-ESM-1-2-LR uses FESOM to drive the PICO box model developed by Reese et al. (2018), which is built into PISM and paramaterizes ocean/ice shelf interaction as well as sub-shelf circulation.

    The PISM setup presented in this study simulates the GrIS, and runs on a polar stereographic grid with a spatial resolution of $5 \, \mathrm{km}$ (geographic coordinate system WGS 84, EPSG code: 3413 (https://epsg.io/3413)), following the newest ISMIP6 standard. As a topography boundary condition for the simulation of ice sheets we use the BedMachine v3 bedrock topography

data set (Morlighem et al., 2017). While this study focuses on the interaction between climate and the GrIS, any other past and present ice sheet domains are ignored. Further ice sheet domains for Antarctica and the entire Northern Hemisphere have also been implemented.

## 2.3   Scope: The Standalone Coupler

The coupling interface — SCOPE, the **S**tandalone **CO**u**P**l**E**r — facilitates coupling between the AWI-ESM-1-1 climate model

component and the PISM component. It is a standalone Python program, operating sequentially between individual short term integrations ("chunks") of each of the two model components. The coupler is modular. Interfaces to both ice sheet and climate components are implemented in such a way that arbitrary models can be connected to each other, irrespective of whether or not they provide coupling-relevant information in a format that fulfills requirements by the respective other model component.



Thereby, SCOPE introduces a generic layer which can be reused when introducing a model, that has been previously coupled via SCOPE, to a different climate/ice sheet model setup. A schematic of this setup is shown in 2.

In contrast to other coupling systems, such as OASIS (Valcke, 2013) or MeSSY (Garny et al., 2019), no actual modification to each model's source code is required to connect two model systems. Coupling is divided into two phases. In the first phase the first model (henceforth the sending model) provides relevant coupling information to the generic layer. This does not only include actual output fields, but also contains a description of the metadata that is relevant for appropriate interpretation and processing by the receiving model, such as variable names, units, and the simulation grid or mesh. The model information is anonymized such that any information hinting at the field origins are removed, e.g. ECHAM6 information is simply referred to as "atmosphere" forcing, or PISM information is simply referred to as "ice" forcing.

After this step has been completed, in the second phase SCOPE is responsible for appropriately processing the data for suitable consumption by the receiving model. Per design SCOPE does not necessitate any modification to a particular model's source code as long as a suitable interface for external reading of forcing data exists. Consequently, the receiving model can read the newly provided forcing data either during restart or during the run itself. As typical general circulation models include such an interface, connecting SCOPE to a new model component does not impose great technical challenges. A standard workflow for receiving data is therefore divided into 3 parts:

1. regridding the data to fit onto the receiving model's grid

2. performing any variable manipulations or corrections, including unit corrections or name changes

3. setting up the model-internal interfaces to actually read the transformed fields; thereby enabling communication between the receiving model and the newly generated forcing data

SCOPE is configured via YAML files, allowing for a separation between user requirements and actual code logic. An example configuration file (Listing 1) for coupling between ECHAM6 (sending) and PISM (receiving) is shown below. Full examples of the coupling within the entire AWI-ESM-1-1 and PISM coupled model system are available with the source code, please refer to the statement on code availability.

## 2.4 Computational Expense

Naturally, model throughput strongly depends on the machine on which the simulation runs and on the employed node configuration. Within the limits of parallel scalability, to some extent one can enhance throughput by employing more computing nodes, which leads to a trade-off in increased overall computational expense. For our model setup, which have balanced to provide a compromise between throughput and computational expense, we derive a typical integration wall time for one model year (computed on `compute` nodes (2x 12-core Intel Xeon E5-2680 v3 (Haswell) @ 2.5 GHz) OR `compute2` nodes (2x 18-core Intel Xeon E5-2695 v4 (Broadwell) @ 2.1 GHz) "mistral" at the DKRZ in Hamburg, Germany) of $\approx 70\,\text{minutes}$ (88% of integration time) for the climate system, $\approx 5\,\text{minutes}$ (6% of integration time) for the coupling procedures, and $\approx 5\,\text{minutes}$ (6% of integration time) for the ice sheet model, leading to a throughput of approximately 20 simulated years per day, ne-





---

**Algorithm 1** Example configuration for SCOPE for ECHAM6/PISM

```
 template_replacements:
 EXP_ID: "PI_1x10"
 DATE_PATTERN: "[0-9]{6}"
 scope:
 couple_dir: "/this/directory/points/to/a/couple/location"
 number openMP processes: 8
 echam:
type: atmosphere
griddes: T63
outdata_dir: "/this/directory/points/to/echam/outdata/storage"
code table: "echam6"
pre_preprocess:
program: "echo \"hello from pre_preprocess. Do you know: $(( 7 * 6 )) is the answer!\""
send:
ice:
temp2:
files:
pattern: "{{ EXP_ID }}_echam6_echam_{{ DATE_PATTERN }}.grb"
take:
newest: 12
code table: "echam6"
aprl:
files:
dir: "/this/directory/points/to/aprl/storage"
pattern: "{{ EXP_ID }}_echam6_echam_{{ DATE_PATTERN }}.grb"
take:
newest: 12
code table: "/this/path/points/to/a/grb/code/table"
aprc:
files:
dir: "/this/path/points/to/aprc/files"
pattern: "{{ EXP_ID }}_echam6_echam_{{ DATE_PATTERN }}.grb"
take:
newest: 12
pism:
type: ice
griddes: ice.griddes
recieve:
atmosphere:
temp2:
interp: bil
transformation:
- expr: "air_temp=temp2-273.15"
ocean:
send:
atmosphere:
ocean:
```

---



glecting queuing times. These metrics show that, from the viewpoint of computational cost, adding interactive ice sheets to an AWI-ESM-1-1 simulation does not lead to drastically increasing expense. The models can also be run asynchronously, leading to a much higher throughput of the ice sheet system, although this would compromise several feedbacks and violate the laws of physics. In particular, if the ice sheet model is run for more years than the climate component, the conservation of mass is

violated, and instead only the average mass fluxes are conserved. Given the nature of the scientific problem under examination, such a procedure may or may not be warranted.

## 3 Climate/Cryosphere Interactions

In the following section, we describe the interactions between a climate model and an ice sheet model, using our new AWI-ESM-1-2 as an example. Additionally, we note some specific limitations and improvements that still need to be addressed.

### 3.1 Ice Sheet/Atmosphere Coupling

In direction from ice sheet to atmosphere, the coupling procedure is as follows. After a PISM simulation chunk, coupling information is sent back to the atmosphere model. The data conveyed by PISM back to the atmosphere provides changes in surface elevation, surface ablation and runoff, and changes in the ice sheet extent. These fields are suitable to update ice sheet masks and orography in an atmospheric simulation, as well as representing ice sheet related freshwater fluxes in the climate

simulation. The latter are routed within the atmosphere to the ocean via AWI-ESM-1-1's hydrology scheme.

When receiving this information, the ECHAM6 atmospheric model must perform several steps. First, relative changes to the orography and corresponding gravity wave drag parameters are applied corresponding to the height change sent by the ice sheet. Such changes cannot be directly implemented into the atmosphere model's restart files. Hence, new boundary conditions are computed, based on both the climate model state before the ice sheet simulation and the anomaly provided by PISM. The

new boundary condition is prescribed as a target, which the atmosphere model linearly approaches over the next simulated year of the climate simulation chunk. Next, changes in the ice sheet extent are used to update the glacial mask, which is shared between ECHAM6 and JSBACH. A thickness cutoff of $5\,\mathrm{m}$ of ice is used to determine areas which should be glaciated. As the JSBACH simulation also calculates soil layer moisture, the amount of soil moisture beneath a newly advanced ice sheet is stored and shielded of from the hydrological cycle; should the glacier retreat again, this water can be restored to maintain long-

term conservation of mass. Vegetation beneath a newly advanced ice sheet is removed. In the case of a retreating ice sheet, the vegetation module initializes deglaciated grid cells with bare soil, and the potential to grow tundra. This facilitates the emergence of plant life should climatic conditions favor vegetation development. Finally, ice mass loss from calving, grounded basal mass loss and surface mass loss is incorporated into the hydrology scheme and transported to the ocean. We note that the transformation of calved ice directly into liquid water is a model simplification, and in reality the freshwater flux from

icebergs would instead be released after over time as the iceberg melts. In the case of ice sheet mass gain, a corresponding amount of water is removed from the hydrology scheme in order to maintain mass conservation in the coupled climate/ice sheet system. Here, we can divide the total amount of precipitation over the ice sheet into two parts. Firstly, that part which can





be used for ice sheet mass gain, and secondly, that part that remains liquid. In a model system without an included dynamic ice sheet component (in our case AWI-ESM-1-1 without PISM), the total precipitation over ice sheets is directly incorporated

into the runoff scheme and transported back to the ocean, thus assuming any glaciated area in JSBACH is in steady state with the climate forcing. However, this neglects the retention potential, i.e. a "buffer" on long time scales, that the ice sheets impose on the hydrological cycle by storing vast amounts of freshwater in the form of frozen ice. As the ice sheet grows, precipitation must be removed from the exchange between the coupled atmosphere-land/ocean system, and as the ice sheet shrinks this discharge must be increased accordingly. To correctly represent this effect, the amount of hydrological discharge is corrected

in JSBACH's hydrology scheme during each coupling interval.

### 3.2 Ocean/Ice Sheet Coupling

In this section, we first describe a SCOPE internal solution to providing mass and energy fluxes to an ice sheet model; and next show how SCOPE can set up ocean forcing for use with PISM internal solutions for ocean forcing. Finally, we specify details that are relevant to the FESOM implementation.

Once received by SCOPE, the oceanic forcing has already been anonymized by the sending process. This generic ocean forcing is regridded, and then processed to provide a forcing for PISM. Here, several options exist. Within SCOPE, a three-equation model following Holland and Jenkins (1999) and Hellmer et al. (2013) can be solved for a vertical average corresponding to water entering the sub-shelf cavity, typically at the bottom of the continental shelf break. This solves the sub-shelf mass and energy balance via the following set of equations:

$$
\begin{align}
\quad T_B &= aS_B + bp_B + c \tag{1} \\
Q_{\text{heat}} &= Q_I - Q_M \tag{2} \\
Q_{\text{brine}} &= \rho_M w_B (S_I - S_B). \tag{3}
\end{align}
$$

The first equation (Eq. 1) computes the melting point temperature ($T_B$) at the ice shelf base/ocean interface via a linearized equation, where $p_B$ is the water pressure at the ice shelf base, and $S_B$ is the salinity of ocean water at the base of the ice shelf.

The parameters $a$ and $b$ represent the salinity and pressure coefficients of freezing equation while $c$ is a small constant offset. In our setup, $a = -5.73 \times 10^{-2}\,^\circ\text{C}\,\text{psu}^{-1}$, $b = 9.32 \times 10^{-2}\,^\circ\text{C}$, and $c = -7.53 \times 10^{-8}\,^\circ\text{C}\,\text{Pa}^{-1}$ (Holland and Jenkins, 1999). The second (Eq. 2) equation describes the flux of heat ($Q_{heat}$) between the heat reservoirs of the ice ($Q_I$) and the ocean mixed layer ($Q_M$) that is in direct contact with the ice base. The mixed boundary layer separates the ice base from the ocean beneath and controls the exchange with the open ocean. The third equation (Eq. 3) specifies the salt flux ($Q_{brine}$) which is proportional

to the basal melting rate ($w_B$), the mixed layers density ($\rho_M$), and salinity difference between ice (in first-order: $S_I \approx 0$) and the underlying ocean ($S_B$). During the coupling, we compute energy and mass fluxes based on the actual ice shelf draft from the ice shelf geometry with the melting rate $w_B$. Here, we utilize the ocean temperature and salinity at the modeled ice shelf draft depth. In our implementation, the calculation uses constant thermal and saline exchange velocities. The computation via the three-equation model provides the basal melting rates and the apparent temperature at the ocean-ice shelf interface. The



latter is essentially the pressure and salinity-dependent freezing temperature (Eq. 1) as the interface is in equilibrium with the surroundings.

Instead of using this to directly provide heat and mass fluxes, SCOPE can also produce fields for the PISM internal parameterizations. If the ocean model is able to provide sub-shelf melting rates and temperatures directly, these can be read into PISM to force melting or refreezing and to drive the basal ice shelf temperature after SCOPE manipulates the variable metadata to

conform to PISM's expectations. Generally, other parameterizations of basal conditions implemented in PISM do not utilize the full three-dimensional structure of the relevant time-dependent hydrographic data (i.e. temperature and salinity). Instead, PISM expects two-dimensional maps of temperature and salinity. Therefore, SCOPE can average hydrographic quantities over a given depth interval, for example $150 - 500\,\mathrm{m}$ in our setup for the Northern Hemisphere. In our case, we use SCOPE to prepare inputs for the PICO ocean box model (Reese et al., 2018), which delivers the basal ice shelf conditions by exploit-

ing these depth-averaged maps. PICO describes the vertical overturning circulation and exchange with the open ocean ("ice pump") based on the three-equation system within the box model of Olbers and Hellmer (2010). The PICO model determines the basal conditions for individual ice shelves for predefined basins.

FESOM provides three-dimensional information regarding temperature and salinity to the ice sheet; however, in contrast to the atmospheric model, the nature of FESOM's unstructured mesh necessitates additional preprocessing in order to enable

regridding on the receiving side of the generic layer to the ice sheet model resolution. Specifically, information regarding the longitude, latitude, and depth are attached to each of the elements' node points, which allows SCOPE's regridding tools to manipulate the forcing. Since the FESOM has static basins, it does not necessarily resolve ice shelves or flooded regions due to a retreating grounding line. The three-dimensional ocean fields are extrapolated laterally into the land. The extrapolation takes into account the spatially nearest simulated ocean value. The vertical averaging and variable manipulation to the ice

sheet model occur similarly for all available ocean boundary conditions. Examples of sub-shelf melting rates provided by this method are also presented in Section 4.2.

## 4  Model Tests

We test our new model by simulating warm climate states described in PMIP4. We choose model setups that correspond to PI-CTRL, MH, and LIG, as described in the introduction. We examine not only the mean climate states, as would be possible

with a less comprehensive atmosphere/biosphere/ocean model, but also examine changes to the ice sheets and illustrate their dynamics, with a particular focus on those quantities that may influence the remainder of the climate, and potentially tip the ice sheet (Notz, 2009) and, subsequently, the coupled cyrosphere/climate system into a new state: ice sheet orography, total ice sheet volume in sea-level equivalent, and ice sheet mass changes resulting from basal and surface melting, as well as iceberg discharge.





## 4.1 Spinup Procedure

In order to produce typical snapshots experiments as normally presented in PMIP, we spinup our coupled climate/cryosphere simulations using a multi-step approach:

1. We spin up the climate model using reconstructed ice sheet geometries and appropriately adjusted orbital parameters and greenhouse gas concentrations. In the simulations presented here, the climate-only spinups are each integrated for more than $3,000$ years.

2. Using the climate state of these simulations, we generate forcings for the ice sheet model. The ice sheet model is initialized from a full glacial cycle index simulation, as described by Niu et al. (2019), and then allowed to equilibrate to the simulated climate state. We consider both atmospheric as well as ocean forcing to spin up the ice sheet by means of standalone ice sheet simulations. The ice sheet-only runs are integrated for $10,000$ years. Here, we judge the equilibrium of the ice sheet by means of examining the sea-level equivalent ice sheet volume.

3. At this point, we consider both climate and ice sheet systems to be in equilibrium with the prescribed orbital and greenhouse gas values that refer to a specific PMIP4 time slice. Ice volumes are stable, and upper ocean salinity and temperature trends are negligible. We now couple both ice sheet and climate systems together in an asynchronous fashion, first running small chunks of the climate model corresponding to 3 simulated model years, followed by 25 model years for the ice sheet model, exchanging coupling information after each chunk. This is to ensure that the climate model can react to changes that the ice sheet has undergone relative to the reconstructed geometries used in step 1, and that the ice sheet in turn can react to any changes in the simulated climate forcing. We do this for at least 1000 ice years, and judge the system to be in equilibrium based upon fluctuations in the mass balance of the ice sheets; which should be close 0 in an equilibrated system. In this case, any changes that may take place in the deep ocean are still in disequilibrium. The coupling interval is asynchronous to allow the ice sheet geometry to quickly adapt to the new climate forcing.

4. Once the models have readjusted to the new geometries, temperatures, and melting rates, we begin to synchronously couple the models, running each system for 3 years before exchanging information.

We show changes in total solar incoming radiation at the top of the atmosphere relative to the Pre-Industrial control experiment for the two paleoclimate snapshots, and corresponding changes in sea-level equivalent ice sheet volume during the ice-sheet only spinups from step 2 in Figure 3. One can note that the changes in ice sheet volume are in line with the expectations from the positive solar forcing anomaly over the GrIS relative to PI-CTRL. Over the entire globe, the summer insolation maximally increases by $\approx +27.5\,\mathrm{W\,m^{-2}}$ for MH in boreal summer, and correspondingly by $\approx +67\,\mathrm{W\,m^{-2}}$ for the LIG, resulting in near-surface air temperature changes. The PDD surface mass balance scheme generates increased ablation. The Pre-Industrial GrIS equilibrates to an ice sheet volume equivalent of approximately $7.4\,\mathrm{m}$ of sea level rise, whereas under constant Mid-Holocene forcing, the GrIS equilibrates at just over $7\,\mathrm{m}$ sea level equivalent volume. For the LIG radiative forcing the coupled AWI-ESM-1-2 generates a total GrIS ice volume that is the equivalent of $6.7\,\mathrm{m}$ of sea level. These simulation





results confirm that the standalone PISM setup, forced with AWI-ESM-1-1-LR climate forcing, is able to reproduce general ice sheet characteristics that are in line with the assumptions made based upon sea level reconstructions (Dutton et al., 2015). In particular, sea level reconstructions suggest a LIG GrIS being smaller than a MH GrIS, which in turn is smaller than a PI ice

sheet.

Next, we examine the simulated ice sheet thicknesses following the ice-sheet-only simulations as these will then be used as new orographic boundary conditions for the subsequent climate simulation (4). We additionally demonstrate the dynamics simulated in our cryosphere simulations by showing the vertically integrated ice velocity. The Pre-Industrial control simulations show similar ice sheet thickness and velocities as are presently observed. The increase in the central ice dome's thickness seen

in both the Mid Holocene and Last Interglacial timeslices can be linked to changes in the SMB, which in our case is primarily driven by precipitation changes. However, the increased melting at the ice margins leads to a net decrease in overall ice volume. Intensified melting occurs in our simulations due to the warmer temperatures simulated during the summer months for both the MH and LIG. Changes in the ice velocity may be related to increased orographic gradients, leading to faster ice flow. The ice divides also shift slightly due to changes in the location of the central ice dome.

## 4.2   Coupling Fields

In this section, we briefly present several examples of forcing sent through the SCOPE, and demonstrate the influence of several of the downscaling and surface mass balance choices provided within the atmosphere/PISM interface. We also discuss the influence of these choices on the ice sheet's surface mass balance.

Under a control simulation representative of the Pre-Industrial, we test our model's ability to simulate the surface mass

balance of the GrIS. Shown in Figure 5 are the inputs for the PDD mass balance scheme 2 meter (a) surface air temperature and (b) total precipitation, both as yearly averages. We then show the integrated surface mass balance over the whole year produced by PISM's PDD scheme (c). The accumulation zone in the southeast of Greenland is well represented by the model, whereas the interior of the ice sheet receives very little snowfall; which is similar to observations. A detailed analysis of the suitability of our PDD scheme can be found in Krebs-Kanzow et al. (2018a) and Krebs-Kanzow et al. (2018b).

Next, we examine the inputs and outputs to the ocean interaction component between FESOM and PISM, the PICO box model. Shown in Figure 6 are temperature and salinity for the control state, as well as the corresponding melt rates as computed by PICO. Note that the values have already been interpolated to the PISM grid, and that FESOM offers a spatial resolution of between $20 - 30\,\mathrm{km}$ around the Greenland coast; in this case the mismatch between the grids of ocean and ice sheet is not as severe as between the atmosphere and ice sheet models. Nevertheless, since the coastline representation between the two

models may differ, the ocean forcing data is extrapolated via a nearest-neighbor algorithm after interpolating it to the PISM computational grid in order to provide forcing to the ice sheet where FESOM might not simulate values due to inconsistency between land sea and ice masks. Our model simulates floating ice in north west Greenland, central west Greenland, and north East Greenland. Given the water mass properties simulated by FESOM, PICO produces melt rates ranging from several centimeters up to $9\,\mathrm{m\,year^{-1}}$.





## 4.3 Applications: Potential for Paleoclimate Research

Given the significant interactions between the cryosphere and the remainder of the climate system, the development of AWI-
ESM-1-2 represents a major step towards producing more comprehensive simulations of Earth's past and future climate. In
this section, we present preliminary results for the Pre-Industrial and compare these against the PMIP timeslices for MH and
LIG; both of which are, time periods in Earth's history where paleoclimate proxies indicate warmer-than-present temperatures
(Otto-Bliesner et al., 2020; Brierley et al., 2020b). As opposed to the coupling tests shown above, these simulations are fully,
bi-directionally coupled. First, we show the accelerated simulations during which PISM and AWI-CM run asyncronously,
followed by results for the synchronously coupled experiments.

## 4.4 Asynchronous Experiments

The asynchronous runs serve to allow the ice sheet to adapt to the climate forcing, which after re-coupling adjusts slightly,
relative to the stand-alone case, as the standalone models do not interact. During the long ice sheet-only spinups from the
viewpoint of the ice sheet the climate state is fixed. Consequently, the ice sheet model does not experience any of the tran-
sient feedback-driven interannual and interdecadal variability that is an inherent characteristic of the climate system. As such,
immediately after coupling, the mass balance of the ice sheet, and, as a result, the ice sheet geometry, adjusts to the changed
forcing/boundary conditions. Furthermore, the climate model still needs to adjust to this new ice sheet geometry. In order to
speed up model equilibration, we allow this adjustment to take place in an accelerated manner. Time series of ice sheet mass
balance and ice volume are shown in Figure 7 for each of the paleoclimate experiments during the accelerated coupling runs.
Once the ice sheet geometry has re-stabilized, we synchronously couple the models together for a further 100 years, which are
used for evaluation of the quasi-equilibrated coupled climate/ice sheet system.

## 4.5 Synchronous Experiments

During the synchronous experiments, each 3 years of the climate simulation are used to force 3 years of the ice sheet model.
After that coupling information is exchanged. We find that the globally averaged near surface air temperature change only
slightly relative to a control case without interactive ice sheets for each of the paleoclimate experiments (Figure 8). For the
paleoclimate experiments, seasonal anomalies for near-surface air temperature, precipitation, sea surface temperature, and sea
surface salinity are shown in Figure 9 for the MH experiment and in Figure 10 for LIG.

The globally averaged timeseries demonstrates that our new model AWI-ESM-1-2-LR, that includes interactive ice sheets,
performs similarly to the simulation with prescribed static ice sheets for the simulations representing the PI-CTRL case. Hence,
the integration of an ice sheet does not degrade the model performance. However, the strongest temperature changes relative
to the PI-CTRL simulation are seen in the DECK simulations for $1\%$ $CO_2$ increase as well as in the abrupt $4CO_2$ simulation.
Globally averaged 2m temperatures increase from $13°C$ to approximately $21°C$ and $19°C$ by the middle of the century. The
simulations for the past warm climates show slight cooling during the Mid Holocene and Last Interglacial (from a globally
averaged mean of $13°C$ to $12.5°C$ and $12.8°C$, respectively), which is in line with slightly lower concentrations of greenhouse





gases. The strongest climatological change occurs for both of these two periods seasonally, which is expected due to the changed orbital configuration. For the Mid Holocene, winters are on average colder; with local cooling of $-2.10°C$ over the Northern Hemisphere high latitudes (field average north of $60°N$). Summer values are comparable to the PI-CTRL simulation.

For the LIG simulation, summers are warmer in the Northern Hemisphere, with temperature anomalies up to $+7°C$. Localized cooling is seen over the Sahara, which is likely due to a shift in the ITCZ and an increase in precipitation, leading to a change in the vegetation coverage. Similar results have also been seen for other simulations of the LIG (Scussolini et al., 2019). Overall, our experiments are comparable to the PMIP simulations for standard atmosphere/ocean models (Otto-Bliesner et al., 2020), and inclusion of the ice sheets does not strongly influence the results. This warrants on the one hand that our coupled climate-

ice sheet model AWI-ESM-1-2-LR does not have substantial biases with regard to other climate-only models for those time periods analysed in the framework of this study, while on the other hand the study of climate/ice sheet interactions is enabled by our model.

An interesting finding here is the clear discrepancy we simulate between the LIG and MH climates. While during both time periods the insolation forcing would suggest warmer than PI-CTRL Northern Hemisphere temperatures during summer, this is

only found for the LIG simulation. During the Holocene, our model produces colder winter temperatures (seasonally averaged 2-m air temperatures), the effect of which is still seen in the summer, likely due to a lag in response. A similar feature is also seen in the sea surface temperatures, with a pronounced cooling of the annual climatologically averaged SST by $-0.5°C$ in the Atlantic, and by approximately $-1.0°C$ in the Pacific. Nevertheless, the simulated climate is similar to that produced by AWI-ESM-1-1-LR simulations without ice sheet interactions as shown in the PMIP ensemble (Otto-Bliesner et al., 2020).

## 5 Conclusions

In the present study, we have presented the new comprehensive Earth System Model AWI-ESM-1-2-LR, comprising of ECHAM6/JSBACH 6.3.01p4/3.11, FESOM 1.4, PISM 1.1.4, the OASIS3-mct coupler, and SCOPE, our newly developed, offline, script based coupling system. We showed the various coupling strategies implemented in our system. In particular, our model allows for a choice of several mass balance schemes; a Positive-Degree-Day approach can be used, or a more so-

phisticated model which implicitly resolves the diurnal cycle via a parameterization, the dEBM (Krebs-Kanzow et al., 2018b). Additionally, our model is able to represent climate/ice sheet dynamics resulting from ice-shelf/ocean interaction. We tested our model with various DECK simulations described in the upcoming IPCC AR6 (CMIP6 DECK simulations), as well as two paleoclimate time slices used in PMIP4. In both sets of experiments, our results agree well with the respective ensemble mean, highlighting that ice sheet interaction does not deteriorate the mean climate state. Hence we expect, that common time

slice climate state simulations deliver similar results. However, since the interaction between ice sheets and the climate system in large can either dampen or amplify certain climate trends, results of transient simulations will probably differ in terms of timing. Our system is now able to resolve several crucial feedbacks which are believed to be of key importance when exploring either transient or equilibrium climate states among all climate components, as it has been shown that both freshwater influence (Barker et al., 2015) as well as orographic changes play a decisive role in the North Atlantic climate in other modelling studies.



A primary limitation of the currently implemented modelling framework is the inability to dynamically adapt coastlines, and as a result, our system can only be used to prognostically understand possible sea level changes. However, future versions of the model will resolve this limitation, thus allowing the exploration of a range of scientific questions on glacial/interglacial to tectonic timescales as well as for the near-term to long-term future. The inclusion of interactive ice sheets into the model Earth System represents a fundamental improvement of the process-representation in simulations, and thus is key in advancing our

understanding of climate/ice sheet interactions as well as providing a first step towards the next generation of fully interactive Earth System Models. Beyond the improvements presented here, several key features would be beneficial to be implemented in the future. Of particular interest for resolving climate-carbon cycle feedbacks during glacial-interglacial cycles is a fully interactive carbon cycle including biogeochemistry. Furthermore, incorporation of iceberg calving and feedbacks with the remainder of the ocean circulation would improve the representation of the evolution of the Earth system, both in the past as well

as in the future.

*Code availability.* Code for the SCOPE coupler is available under https://gitlab.awi.de/pgierz/scope, and can also be installed via the Python package managing system, pip, via the command `pip install scope-coupler`. SCOPE is free software, licensed under the GNU Public License, v.3. The Parallel Ice Sheet Model PISM is available under https://github.com/pism/pism. FESOM is also free software, and available under https://github.com/fesom/fesom. ECHAM6, which is the atmosphere model of the MPI-ESM, is property of the Max Planck

Institute for Meteorology. It's model code is available at https://code.mpimet.mpg.de/login?back_url=https%3A%2F%2Fcode.mpimet.mpg.de%2Fprojects%2Fmpi-esm-users%2Ffiles after registration at https://www.mpimet.mpg.de/en/science/models/availability-licenses. For further information, please contant karl-hermann.wieners@mpimet.mpg.de. Modifications required to reconstruct the version of ECHAM6 in the AWI-ESM are avaiable as patch file under https://gitlab.awi.de/pgierz/echam6-patch.

*Data availability.* Simulation output is provided via PANGAEA (https://www.pangaea.de/), and access information will be provided upon

manuscript publication.

*Author contributions.* PG and CR designed and programmed the coupling interface, with support from CS. PG and DB developed infrastructure suitable for running the simulations on various supercomputers. LA and PG designed the experiments, LA performed the simulations. PG analyzed and visualized the results. All authors contributed to the manuscript and the discussion of the results.

*Competing interests.* The authors declare that they have no conflict of interest.





*Acknowledgements.* PG is funded by the Federal Ministry for Education and Research initiative PalMod: Simulating a Full Glacial Cycle; BMBF grant 01LP1503B (project PalMod1.2). CR has been financed through the German Federal Ministry of Education and Research (Bundesministerium für Bildung und Forschung: BMBF) project ZUWEISS (grant agreement 01LS1612A). GL acknowledges funding via the Alfred Wegener Institute's research programme PACES2. CS acknowledges funding by the Helmholtz Climate Initiative REKLIM and the Alfred Wegener Institute's research programme PACES2. The Deutsches Klimarechenzentrum (DKRZ) supplied computer resources on
the cluster "mistral". All the coauthors would also like to thank the AWI's HPC support team for their proactive and generous assistance enabling this work during the development phase. The development of PISM is supported by NASA grant NNX17AG65G and NSF grants PLR-1603799 and PLR-1644277. This simulations presented in this study were performed using the `esm-tools` (Barbi et al., 2020).

In particular, we also would like to thank colleagues at the Potsdam Institute for Climate Impact Research, Ricarda Winklemann and Ronja Reese, for assisting in setting up the PICO ocean box model.

The data analyzes and the production of figures have been predominantly performed with the help of the following software products (alphabetic order):

– Climate Data Operators (CDO): https://code.mpimet.mpg.de/projects/cdo/

– Generic Mapping Tools (GMT): https://www.generic-mapping-tools.org/

– Ncview: http://meteora.ucsd.edu/~pierce/ncview_home_page.html

– netCDF Operator (NCO): http://nco.sourceforge.net/

– PyFerret: https://ferret.pmel.noaa.gov/Ferret/documentation/pyferret

– Python3: https://www.python.org/, including the following packages

– NumpPy: https://numpy.org

– matplotlib: https://matplotlib.org

– xarray: https://xarray.pydata.org/

We thank the numerous authors and their financial supporters of these software products.



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



645 **Tables**

**Table 1.** Vegetation Types considered in JSBACH

| Vegetation Type | Classification | Type |
| --- | --- | --- |
| 1 | Tropical Evergreen Forest | Forest |
| 2 | Tropical Deciduous Forest | Forest |
| 3 | Temperate Evergreen Forest | Forest |
| 4 | Temperate Deciduous Forest | Forest |
| 5 | Raingreen Shrubs | Grass |
| 6 | Cold Shrubs (Tundra) | Grass |
| 7 | C3 Perennial Grass | Grass |
| 8 | C4 Perennial Grass | Grass |





**Figures**

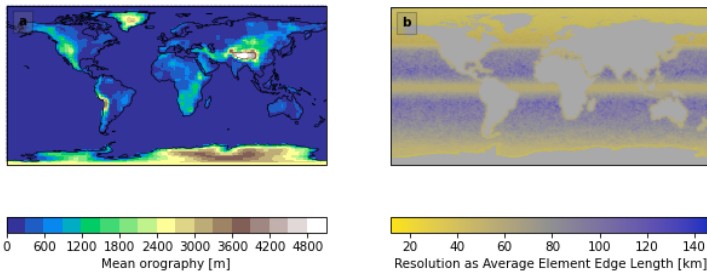

**Figure 1.** Model Resolution. Panel (a) shows the orography boundary condition used in ECHAM6 as an example of the model resolution, and (b) shows the resolution used for FESOM, defined as the average edge length of one triangular surface element.





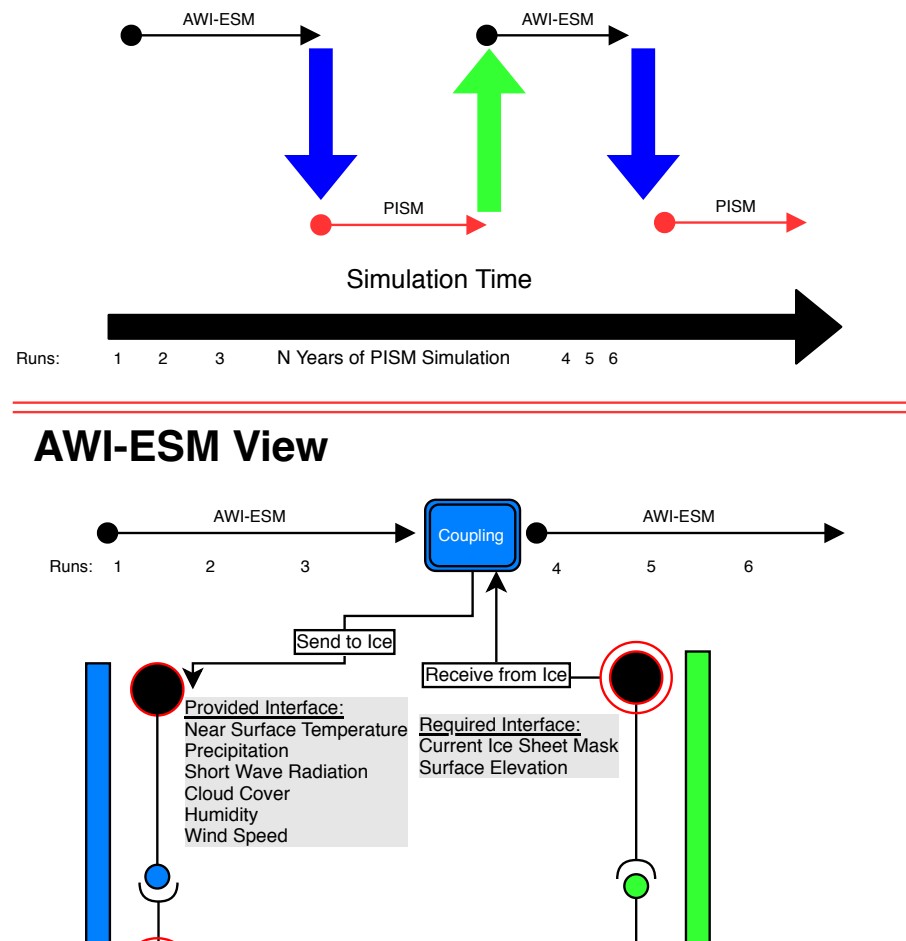

**Figure 2.** Schematic of AWI-ESM-1-2. The upper panel shows the two models running iteratively; whereas the bottom panel depicts the SCOPE coupling procedures.



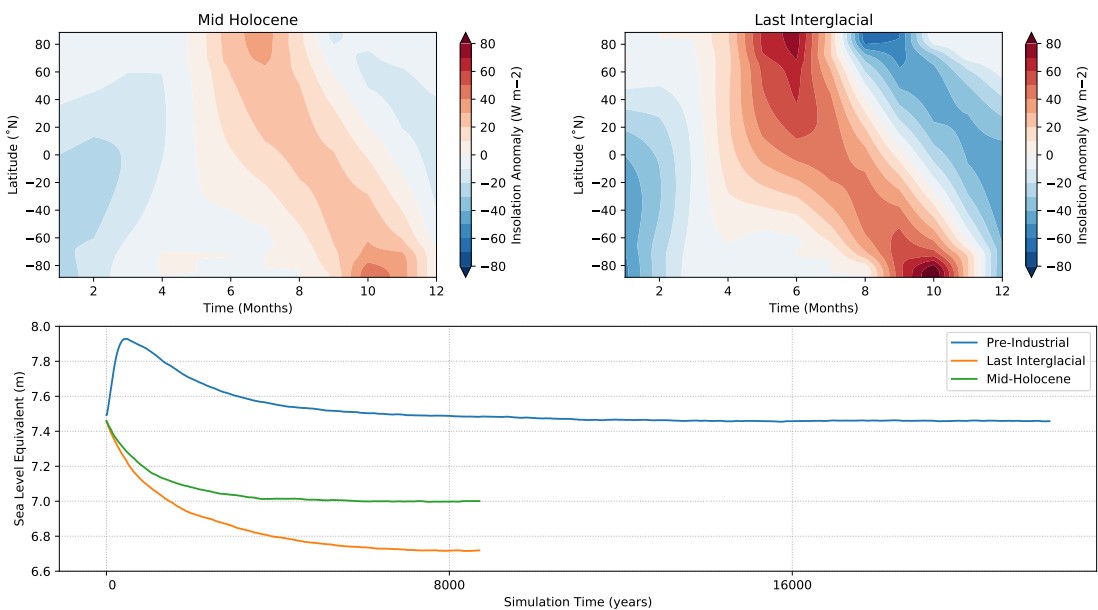

**Figure 3.** Insolation anomalies for **a** the Mid Holocene and **b** the Last Interglacial. Panel **c** shows sea-level equivalent ice sheet volume for the simulated domain as the ice sheet model equilibrates under constant climate forcing, after being initialized from a simulation of the last glacial cycle.

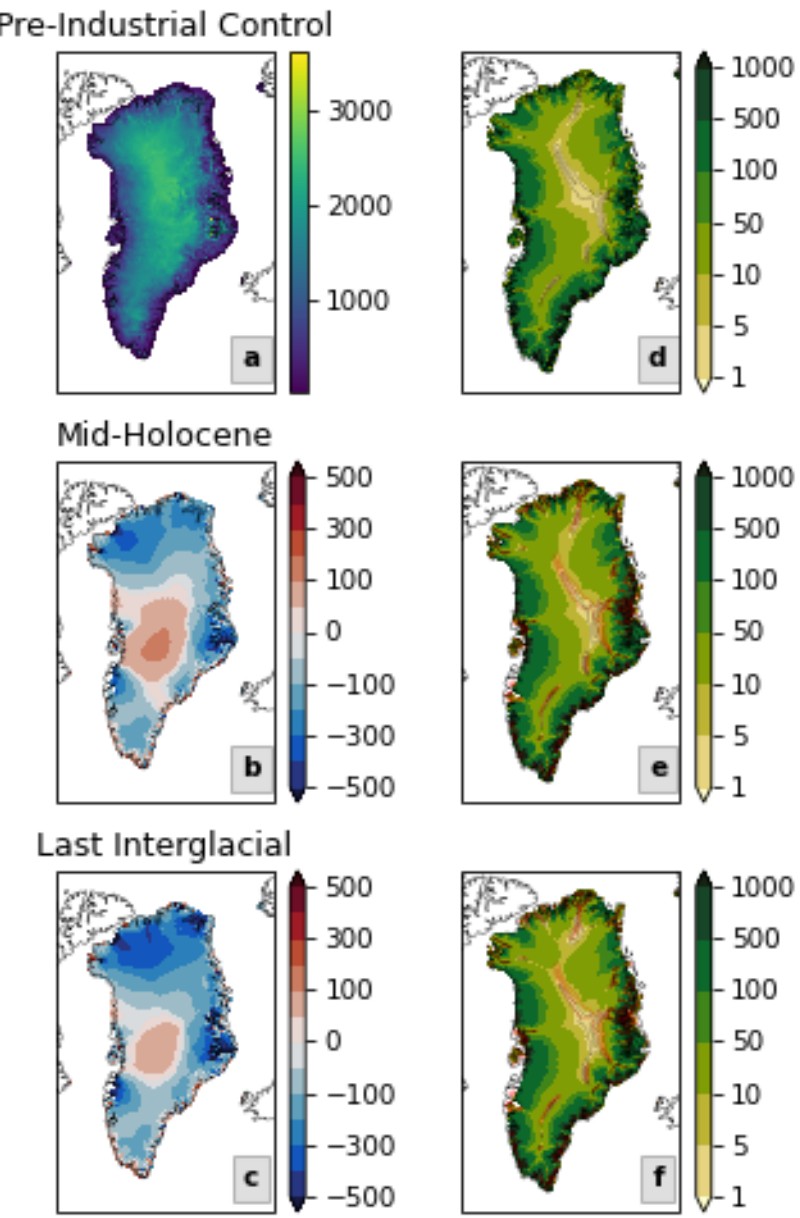

**Figure 4.** Simulated GrIS thicknesses for Pre-Industrial (a) and anomalies for Mid Holocene (b) and the Last Interglacial (c). Panels d, e, and f show the absolute magnitude of ice velocity and the ice divide (area where ice flow is less than $3\,\mathrm{m\,a^{-1}}$) as a black contour. For (e) and (f), the red contour marks the Pre-Industrial velocities for comparison. For the velocity plots, values with surface elevations below $500\,\mathrm{m}$ .





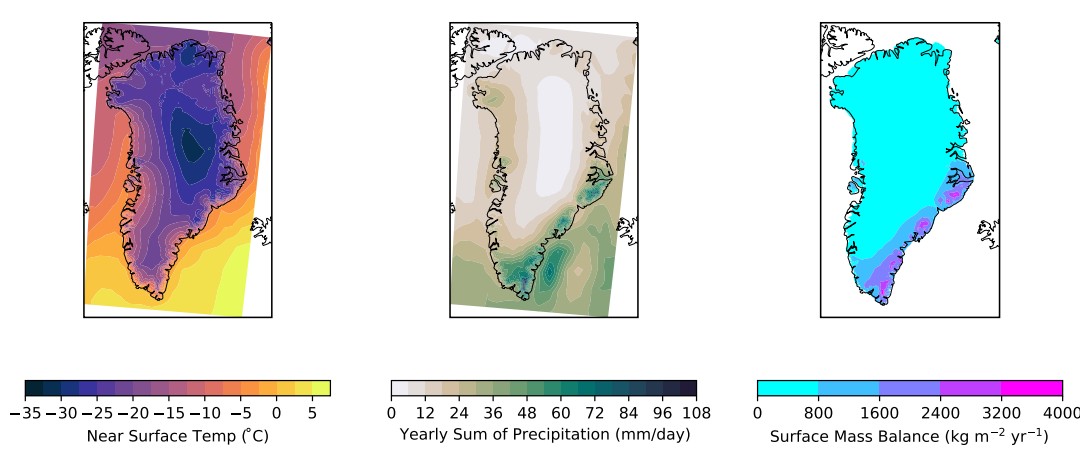

**Figure 5.** Coupling fields sent from ECHAM6 to PISM to accommodate surface mass balance simulations (a) and (b), and resulting surface mass balance for the GrIS (c). Note that in panels (a) and (b) the entire simulated domain is shown, whereas for (c), we restrict the results to only show those areas of PISM which have active ice dynamics, defined as the area where ice thickness exceeds 5 meters.



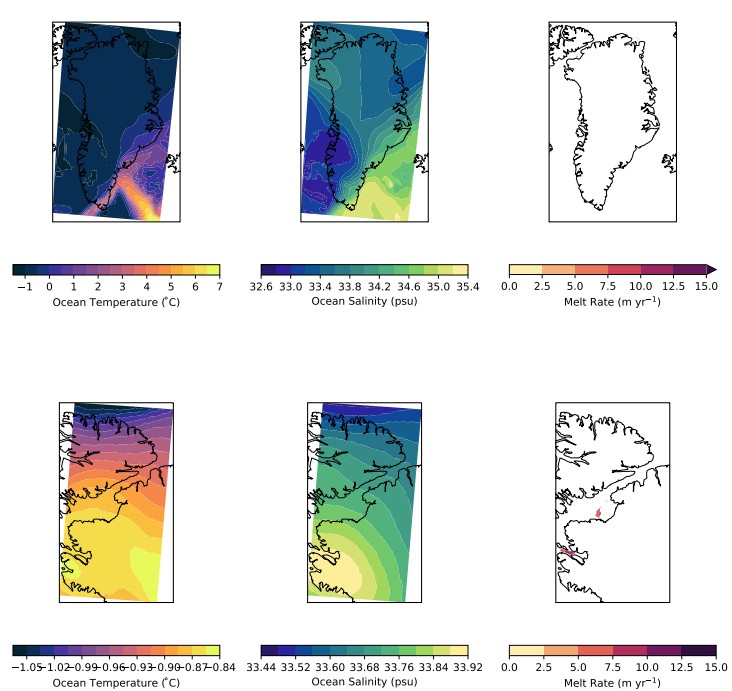

**Figure 6.** Coupling fields sent from FESOM to PISM's PICO box model. (a), (b), (d), and (e) show temperature and salinity inputs for the entire model domain, and a zoomed-in selection of northwestern Greenland. (c) and (f) show the resulting melt rates as calculated by PICO.



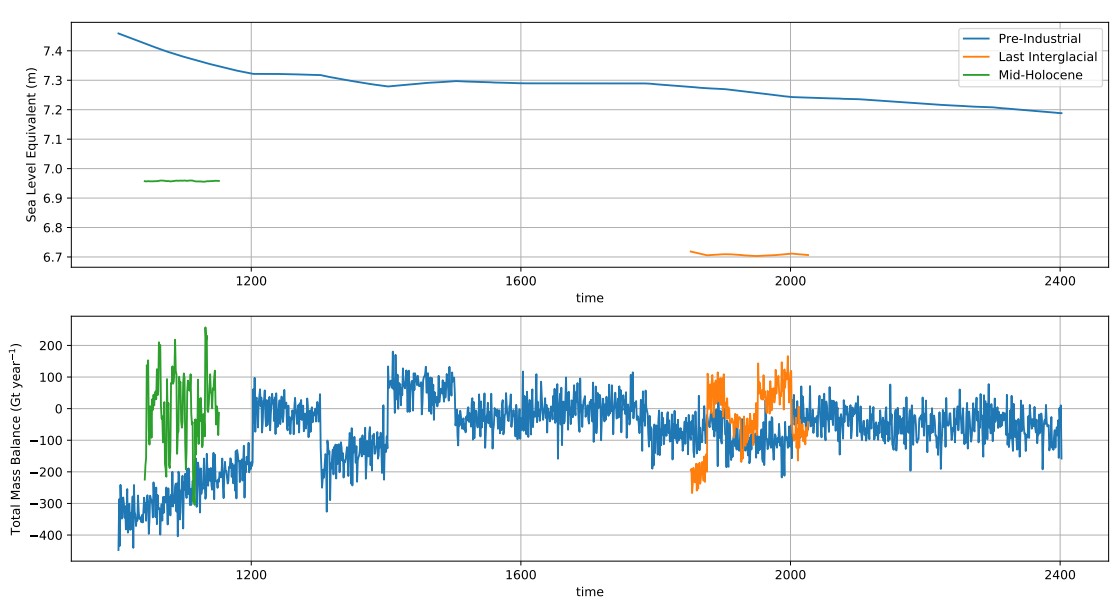

**Figure 7.** Sea-level equivalent ice volume (a) for the asynchronous coupling. The lower panel (b) shows total ice sheet mass balance.



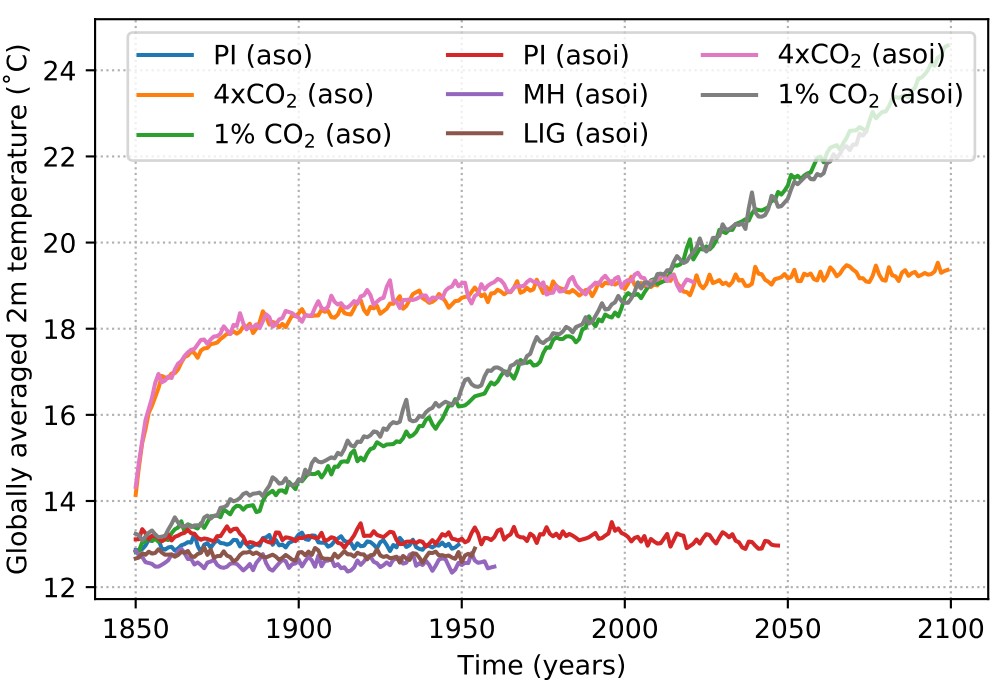

**Figure 8.** Near surface (2m) air temperatures for the suite of model experiments presented in this study. For the DECK simulations, a comparison is made between simulations with and without dynamic ice sheets; and only small changes can be observed.



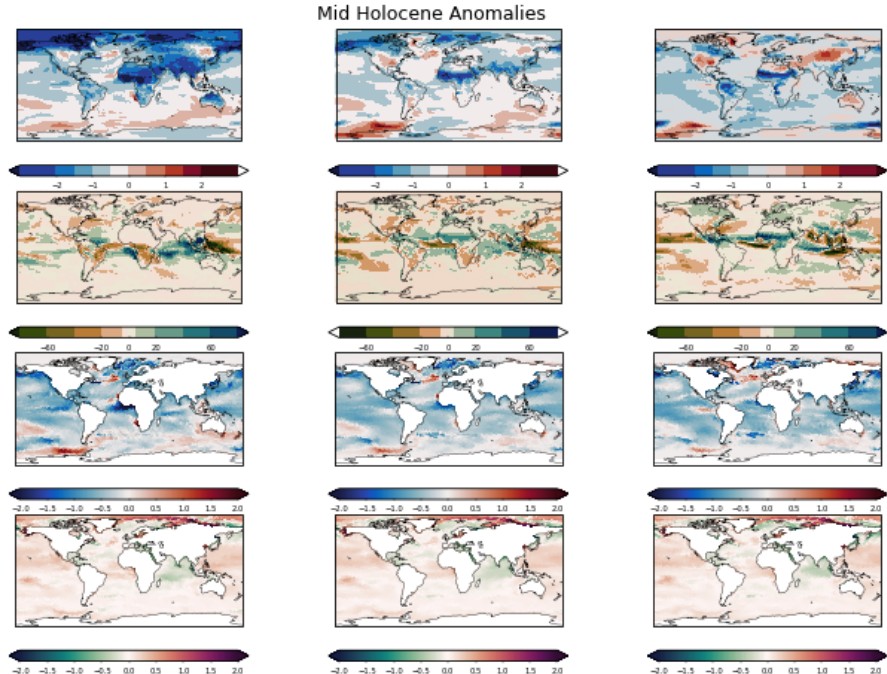

**Figure 9.** Near surface (2m) air temperatures, precipitation, sea surface temperature, and sea surface salinity anomalies for the Mid Holocene experiment. The left column shows winter values (DJF), middle column shows climatological means, and the right column shows summer values (JJA)



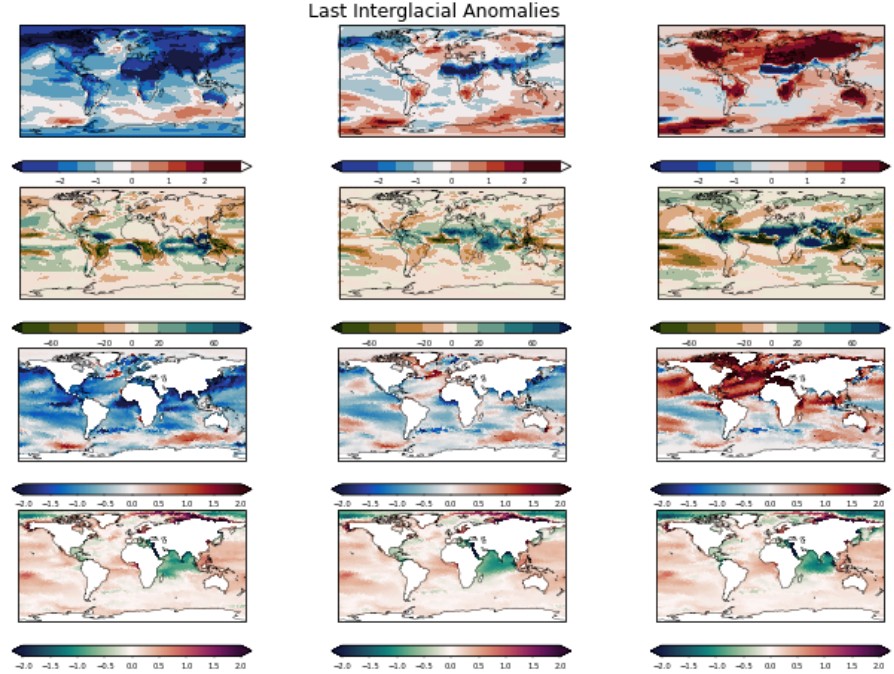

**Figure 10.** As in Figure 9, but for Last Interglacial anomalies.