# Peer review of "Simulating interactive ice sheets in the multi-resolution AWI-ESM 1.2: A case study using SCOPE 1.0"

_Geoscientific Model Development, 2020_

## Referee Comment (RC1) · Anonymous Referee #1 · 6 Sep 2020

*Summary

This paper describes aspects of the AWI-ESM-1.2 climate model, focussing specifically on the SCOPE system used to provide coupling to an interactive ice sheet component. It also gives some general information about the performance of the system in a set of case-studies. As the authors note, there is a pressing need to include ice sheet and shelf couplings to help answer many important questions in climate science, and this work addresses a timely and important topic. The SCOPE coupler at the heart of it all seems to be a flexible and useful tool likely to be of interest to workers in the field. Given the potential abilities of the model and coupler, and the range of simulations that have been conducted, I found overall that this manuscript provides a rather unsatisfying write-up which I think it could be significantly improved for readers by giving more detail

in a number of areas.

*General Comments

Researchers in Earth System modelling are currently working out how best to implement ice sheet couplings in state-of-the-art models, and it's encouraging to see new models like this appearing. The SCOPE framework they describe seems like it is flexible enough to couple between various components, and also couple them at different levels of complexity - eg SMB provided by either PDD-relevant fields or the dEBM - and I think that's a really useful approach that might enable other groups to try plugging an ice sheet into their climate modelling systems. I really like what it looks like they can do, and I'm pleased to see that the authors have included some paleo simulations as well as the DECK in the case studies they've done to illustrate its performance. However, I don't think this manuscript really gives me the level of detail I'd want to see in order to assess how AWI-ESM with ice sheets behaves in practice, nor contains enough to convince other modelling groups that they might want to use the SCOPE coupler. It's really valuable for interested readers to be able to see what the modelling systems do well, and also where they don't, as part of assessing what scientific questions its appropriate to use the model for. As it stands, the paper implies that the coupling system can pretty much do it all, but has only had those couplings tested in a very limited way, and the results of those tests are not evaluated here in much detail.

This is a short text, and pretty much the first third is given over to summarising pre-existing model information. The coupling system is then described in terms that include the potential for a range of ice shelf-ocean interactions, and the Introduction has promised DECK and PMIP case-studies, so it's disappointing that the only details we get about how this all really plays out in practice concern Greenland-only runs in warm climates whose results are sketched out in the barest detail. As a GMD paper this isn't necessarily the place for a detailed description of simulation results for their own sake, but I think it's important that descriptions of a modelling technique come with enough illustration of how they work (and maybe how they don't) to be able to judge whether

the proposed technique/model is useful for a particular application.

I may be misreading it, but the paper feels confused as to whether its purpose is to document how SCOPE and the coupling work in principle, or to describe how AWI-ESM behaves once equipped with ice sheets. Depending on the authors' goal, I might recommend adding different types of information to the paper. In either case, it wouldn't hurt to have a more focussed statement of the scope and aims of the work up front. If SCOPE and the technical adjustments made to the models during coupling are the main focus - perhaps with a goal to advertise SCOPE to other groups - then I think they would improve the paper by providing more explicit sample configuration files that illustrate the flexibility in how it can be used and what transformations and parameterisations SCOPE can take care of and what would need to be altered inside the host atmosphere/land/ocean models. More illustrations of how SCOPE transforms input fields into information suitable for use in the target models for a range of case studies would be good here too. If instead the coupler performance really is inseparable from the AWI-ESM climate simulations it's embedded in, then it would be good to see a lot more /evaluation/ of what happens in those simulations, both to the ice and relevant climate fields - eg not just ice extents with respect to other work, but deep water formation in the ocean and how that affects large-scale heat transports. In an ideal world their case-studies would include some cold paleo climates and other ice sheets as well, to show a broader range of behaviour and sensitivity, but that is probably too much additional work to suggest at this point. Few details are shown in the case study simulations that are presented here, and there is almost no critical evaluation or comparison with other studies in what is presented, which is a little disappointing.

*Detailed comments

abstract: most of the abstract is concerned with motivating the inclusion of coupled ice sheets in climate models, rather than summarising the work that has been done here.

line 5 (and elsewhere): the term "comprehensive" is used in a few places in an abso-

lute sense, but no climate model includes representations of every possible - or even every relevant - physical process. This and "fully coupled" are simply inaccurate terms, unless used relatively to say that a model includes more couplings than others.

line 6 (and elsewhere): "cryosphere" is used in a few places in a way that makes most sense if taken to mean land-based ice, but the term is wider than that - snow cover and sea-ice are part of the cryosphere, for example, and their interactions are common features of most climate models

line 30: repeat of "large scale"

line 40, 618: citation of Shi et al has year 0

line 53: having motivated the need for global ice sheet modelling, this is the first suggestion, almost off-hand, that only Greenland will be looked at in this work. This limitation is only actually made explicit on line 145 at the end of section 2.2 - I think it needs to be said much more clearly, much earlier, in the descriptions of what has been done in the abstract and introduction. Given that the system is apparently technically capable of looking at other ice sheets too, at least that's how I interpret line 146, I think that looking at only Greenland is a disappointingly limited case study. If more ambitious runs had been undertaken I guess that they would have been used instead, but it would have been great to see.

line 58: by "AR4 IPCC scenarios" do they mean the SRES scenarios (Nakicenovic 2000) - that might be a more general and helpful term here?

line 130: Gregory et al 2012 found that the size of artificial variability added to PDD melt calculations can have a major impact on the results - can more detail be given here than simply "white"? Gregory et al, "Modelling large-scale ice-sheet–climate interactions following glacial inception", Clim. Past, 8, 1565–1580, 2012

line 161: "hinting" is a rather informal term. Why does the origin of the data have to be anonymised completely, it seems that having it potenitally retained in some form of

metadata might be useful in some cases if SCOPE can be used to generate boundary condition files from different source models that could in principle be archived for later use?

line 186: I don't disagree with the conclusion that the ice sheets are likely not a major additional computational expense in the context of current CMIP ESMs, but it perhaps should be noted that the runs here only include Greenland at 5km. I imagine an Antarctic sheet/shelf system would cost significantly more, as would glaciated areas appropriate to a Last Glacial Maximum scenario, which I imagine is in scope since PMIP is an area of interest here. That thought raises another question - how would their system run separate GrIS and Antarctic ice sheets? In one global domain, or totally separate instances of PISM? Would that require separate SCOPE instances too? The timesteps and coupling intervals being used in the system have also not been cited, which might be useful.

line 188: "violate the laws of physics" sounds extreme. I think outlining specific conservation issues implied by the asynchroneity would be better.

section 3.1: I'm not really clear on a couple of issues related to melt and conservation that run through this section. Perhaps they could be explained more clearly. I appreciate this stuff is tricky to explain - my description of my confusion that follows is not the clearest either, but should gave a flavour of the aspects I think could be rephrased.

As ECHAM/JSBACH runs, it must make some estimate of the melt over the ice sheet, if only as part of the latent heat calculation and estimating surface temperature. Is melt actually produced, and routed to the ocean, or is there no explicit melt/runoff occurring at all under an assumption (line 220) that the ice sheet (without interactivity) is in steady state so ice sheet runoff in JSBACH is just whatever precipitation has fallen? Under a PDD scheme, PISM will be calculating melt in a wholly different manner, unconnected with what happened in JSBACH - so that must be why the PISM ablation (line 198 - but under a PDD SMB scheme how is surface ablation different from runoff, is there basal

water too?) is handed back and (line 224) the "hydrological discharge is corrected". That year's discharge has already happened in JSBACH though, so is an adjustment made to the amount of the next year's precipitation that makes its way to the ocean? Is that adjustment spread through the year evenly, or with some seasonal pattern? Does it have a spatial pattern? Is any attempt made to reconcile the surface latent heat fluxes with the melting that PISM has seen?

line 205 says the orography changes are phased in over the next year of ECHAM run. Is the same true of the extent/mask changes, or are they instantaneous? In some models, the surface roughness (linked to the glaciated mask) would be related to the subgrid orography terms. Linking to my question above, are the various water fluxes (line 212) phased through the year of climate run too? The discharge correction is not noted here, but it does say that surface mass loss is part of what is transported to the ocean - is this term actually the (corrected) ECHAM precipitation, or really the PISM surface ablation?

Line 206 simply notes that the JSBACH glacial mask is updated. The PISM horizontal resolution is much finer than that in JSBACH - is there a threshold for the amount of a JSBACH box that is glaciated in PISM used for this, or can JSBACH cope with both fractional ice extent in surface type and soil representations? Soil water is buried by advancing ice, but vegetation is destroyed - does JSBACH track carbon or nutrient in the soil, and is this buried as well? Is the vegetation mass/carbon simply lost from the model completely as ice advances, or conserved somehow?

Line 215: calved mass would ideally not only be released over time, but over quite a large area as well - is all of this water placed at the surface, right at the coast?

section 3.2: given that the previous section is so specific about what is done in the ECHAM/JSBACH case, it feel incongruous that this section talks mostly about capabilities that aren't used in AWI-ESM. It also only deals with the provision of a melt rate to the ice shelf, with nothing on information that might be passed back to the ocean, such

as shelf geometry or adjustments for heat or salinity conservation based on meltrates that might have been computed in PISM or PICO. Can this be done in their system - the land-surface equivalent is described in the previous section for ECHAM/JSBACH?

line 230: the first case described seems to be relevant for where the ocean model explicitly resolves shelf profile. Does SCOPE take 2d fields along the shelf surface, or a 3d fields from the basin for this? When regridding, is any adjustment made for the greater resolution of surface geometry detail on the PISM grid, equivalent to the lapse rate adjustment to the surface temperatures in the PDD melt calculation?

line 259: I'm not clear on where PICO sits in the framework here. Is it somehow built into SCOPE, or PISM? Does SCOPE prepare FESOM output for PICO, then get called again to process PICO output for PISM? If so, how are the separate parts called and coordinated - is coordination of sub-models a SCOPE function too?

line 267: further to the note on ocean vs ice sheet model resolution above, surely it's not just that FESOM isn't resolving areas beyond the grounding line, but for the Greenland case-study used for the rest of the paper, it's not resolving /anything/ of the fjord systems that lead up to the marine-ice interfaces that PISM wants boundary conditions for. This is where PICO comes in as not just an option but absolutely essential.

line 273: as noted before, it's a bit disappointing that only warm PMIP climates have been looked at. Could more comment be made on whether AWI-ESM could/will be used to look at colder paleo ice as well? The PI is also a bit tricky in terms of being their most modern case-study, it's much easier to provide an evaluation of the model performance for the present day for which there are actual observations.

line 274: this sentence is a good description of what I think should be in this section, but what is actually written in section 4 doesn't live up to this ambition yet.

line 302: I don't think it's said anywhere why a three year coupling period is chosen. I may have missed this detail, but line 130 notes that monthly data is saved from the

atmosphere model - does the coupling pass a 3 year timeseries of atmospheric forcing data to PISM, or does it construct a climatology by averaging over those three years somehow?

line 317: I think a "Fig" is missing from the number in brackets "(4)". This figure is useful, but it would help the reader to see a more detailed evaluation of the LIG, MH and PI GrIS states compared to other reconstructions / simulations

line 337: The FESOM resolution is still not sufficient to resolve the majority of the coastal system relevant to the ice marine boundary conditions for the outlet glaciers, so although the ice/ocean grid numbers match up better there's still an awful lot of real physics missing. This is inevitable in a global model, but I think it should be stated more clearly. I'm not clear on how exactly PICO is deployed here either - are there individual (and individually tuned) PICO boxes for each outlet glacier, or are the different areas aggregated somehow? How has PICO been tuned for use here? Again, some evaluation of how the melt rates achieved stack up against other evidence, or even simply what is observed in the present day, would be useful.

line 351: typo, "asyncronously"

line 354: are these simulations accelerated with the same protocol as used in part 3 of the "Spin-up Procedure", eg 3 climate years -> 25 ice years?

section 4.5: At less than 1 page, this section is simply too short to usefully illustrate or evaluate the model performance across two climate change scenarios and two paleoclimate simulations. Given that a coupled Greenland ice sheet is the main feature of the model in this paper, it is also a little bizarre that the section focusses on global, large-scale climate fields, with an evaluation limited to the fact that the coupled Greenland doesn't make much difference to the simulations. As it stands, it's little more than a statement of the simple fact that these climate setups can be run with AWI-ESM, it doesn't tell me anything about the ice sheet, how the coupling is working in the model or anything about the climate system in reality.

line 411: I don't think "prognostically" is used correctly here. Not being able to change the coastlines mid-run is an important limitation, common to many models with coupled ice sheets, and should be mentioned earlier when the coupling system is first described.

fig 2: this schematic only shows atm -> ice coupling, and omits the ocean/PICO coupling processes?

fig 4: figure has no units

fig 5,6: caption doesn't note the source (eg simulation, time-averaging etc) for the coupling fields. It might be good to show these as block-fill plots rather than smooth contours, that would illustrate the resolutions that the different models are working at - 5a and b could show the ECHAM grid, and 5c the PISM one, for example.

fig 9,10: figure has no units, caption should note the reference climate for the anomalies (preindustrial)

[Figure]

---

## Referee Comment (RC2) · Anonymous Referee #2 · 13 Sep 2020

General comments:

The focus of the paper has not become clear to me and there are contradicting signals in title, abstract and main text. Some parts read like the paper should be a description paper of the SCOPE coupler. Other parts suggest that it is a case study on fully-coupled simulations of the Greenland ice sheet. At the same time, references to other ice sheets (NHISs, AIS) are present suggesting that the work could be considered as establishing a fully coupled system, ready to be used for any (paleo) configuration. I see considerable shortcomings for the two latter interpretations, which makes me lean to suggesting a specific focus as a SCOPE description paper. In any case, the paper should be considerably reworked to make it clear from title to abstract and introduction what the focus of the paper is.

[Figure]

The manuscripts lacks important references to earlier and similar work on including interactive ice sheets in Earth System Models. At the same time, it seems that the authors do not fully appreciate the complexities associated with such undertaking (see e.g. Fyke et al., 2018). Considerable efforts have been made and are ongoing e.g. to improve the representation of the SMB over ice sheets (e.g. Vizcaíno et al., 2010; Sellevold et al, 2019) and to produce consistent coupled initial states for climate-ice sheet simulations (e.g. Fyke et al., 2014). Shortcomings of the current modelling approach should be critically discussed in view of these and other existing studies (e.g. Smith et al. 2020).

It is mentioned in P6 l147 that other domains are implemented. But why are they not analysed? I don't think they can be considered similar enough so that showing the model for a Greenland case only is sufficient. AIS and NHISs have considerably different characteristics. In particular the interaction with the ocean of these marine ice sheets is clearly a different case than what can be done with a predominantly land-based Greenland during warm periods. The PICO model e.g. has been specifically developed for the Antarctic case. It is unclear to me why it is tested in the Greenland context. The interaction of the ocean with Greenland outlet glaciers is clearly not adequately represented in this model setup, which is a severe shortcoming for this use case that should be discussed.

If the aim of the paper is to show that the coupling is functional beyond a purely technical nature, it is crucial to see some critical experiments that explore the model's capabilities. How does the SMB for the present day over Greenland compare to observations and other model results? Is the SMB anywhere close to adequate as boundary condition for an ice flow model? If not, how does that limit the predictive capability of the model as a tool to look into warm climates of the past and future? How could the SMB be further improved? How does the atmosphere respond to an ice sheet that is considerably lowered and for an ice sheet that is retreating over land? How does the ocean respond to freshwater input? All of these are questions that need to be addressed should the model be used for simulations of the PD, LIG and future. And many more questions arise if the model should be employed for colder periods.

It should be noted somewhere prominently that the GrIS cannot be expected to be in a steady state neither for the LIG nor for the MH. We can of course use snapshot climate simulations to study the effect of a climate perturbation of a certain pattern and magnitude, but it should be clear that we are not looking at a real (transient) climate experiment. Comparing the GrIS sea-level contribution from these experiments with e.g. LIG reconstructions of sea-level is therefore problematic and requires some additional comments.

Title:

From the title, it is not clear what SCOPE 1.0 is. Maybe "A case study using the Standalone Coupler SCOPE 1.0"

The term "multi-resolution" in the title is not picked up in the manuscript. Suggest to remove it or add substance concerning this feature in the manuscript.

The title mentions "ice sheets", but the paper presents only results for one ice sheet (Greenland). Suggest to rephrase.

Abstract:

Important elements from the manuscript and title should be present in the abstract. The coupler SCOPE is not mentioned in the abstract, while it is an important part of the manuscript.

P1 l19 I think you mean "future ... studies" as in "upcoming". Try to avoid paring "future" and "paleoclimate" in this sentence.

Comments:

P2 l25 It seems counter-intuitive to measure the amount of freshwater stored in the ice sheets in SLE. Could give the percentage of the global freshwater supply instead. Also

not clear why the glaciers are included in this point.

P2 L28 While you discuss the AIS, GrIS and glaciers before, here is only information about recent changes of the GrIS. Not clear why. Should extend to the AIS.

p2 l30 Avoid repetition of "large-scale"

P2 l31 What is "interior ocean circulation"? Rephrase?

P2 l36 What are these shifts? Heinrich events, or DO events?

P2 l37 "trigger or response". Neither trigger nor response suggest the notion that you put forward earlier of a fully coupled system with feedbacks. This may be important to reformulate.

P2 l38 "Earth System models with the capability"

P2 L39 Add version number after AWI-ESM?

P2 L40 Remove "that is"

P2 L40 Reference year Shi et al missing.

P2 L42 "which is implemented"

P2 l45 "The simulations are based on"

P2 l48 "time slice simulations". Need to clarify how that relates transient coupled simulations.

P2 l50 "climate states". Why plural?

P2 l51 Ice sheets can always have runoff. Do you mean anomalous runoff.

P2 l53 There is no motivation given why the focus is now solely on the GrIS. Please motivate that choice and why that is a good test case or easier to handle than the AIS or NHISs.

P3 l76 This is a difficult sentence, consider introducing an abbreviation for ECMWF before and reformulate.

P3 l84 Consider introducing abbreviation ECMWF before. See also previous point.

P4 l116 Please describe what the implications of using dynamic vegetation are.

P5 l126 The PDD method is often assumed to be ill-fitted for paleo applications with different orbital parameters. This should be discussed as a possible caveat.

P5 l128 The units of the PDD factors are 3 orders of magnitude wrong. Typically 8 mm d-1 K-1 for ice.

P5 l130 "forcing" is repeated here. Not clear what "forcing adds white noise" means Reformulate.

P5 l132 Sea-proximal. For Greenland this concerns marine-terminating outlet-glaciers. Reformulate?

P5 l133 Are the three parameterisations employed simultaneously? Clarify.

P5 l146 The past sentence of the paragraph contradicts the sentence just before.

P6 l155 "shown in Figure 2"

P6 l162 Could you please explain why the information needs to be anonymized?

p6 l173 YAML needs a reference.

p6 l182 This sentence does not read well and contains too much information.

p6 l185 While the time spent for the coupler takes a relatively small percentage in the entire model, it must be much slower than an online coupling procedure within the model. It would be interesting to read some discussion about that and an estimate how much the coupling time could be reduced if a more efficient procedure would be employed.

P7 This is difficult to read (too small font) and does not provide critical information. Improve and move to the appendix?

P8 L189 It is not clear to me why and it sounds drastic to claim that running asynchronously means a violation of the laws of physics. Reformulate? I seem to understand that you do run the model in this way during initialisation, so this should be mentioned here as a case where it makes sense to employ the model in this way.

P8 l195 I miss a paragraph 3.0 about Atmosphere/Ice sheet coupling. How to produce an adequate SMB as boundary condition for an ice flow model is not at all obvious. The large difference in resolution between the atmosphere model and the ice sheet has to be bridged somehow. The fact that you use the PDD scheme in the ice sheet model does not make it easy to separate this problem, but should be discussed nevertheless. How is the atmospheric information interpolated/ downscaled. Does the PDD calculate in anomaly mode or with the absolute temperature. How do you deal with the fact that atmospheric grid cells can contain a mix of ice sheet, ice-free land and ocean?

P8 l197 Be precise and explicit about what information is passed as change/anomaly and what as absolute field. What is the reference elevation (in the climate model and in the ice sheet model) if changes are communicated? Is the extent really communicated as a change in extent?

P8 l198 Since you distinguish ablation and runoff, which field is used for what? WHat do you use ablation for?

P8 l200 Why is runoff routed by the atmosphere and not by the land surface model as mentioned on p3? Is this discussion not better placed in the interaction with the ocean? How do you distinguish runoff from subglacial discharge and from frontal/sub-shelf melt.

P8 l200 What is the hydrology scheme in AWI-ESM-1-1? Explain.

p8 l216 How does this removal from the hydrological scheme work. Is it globally distributed?

p9 l230 What are the exchanged quantities?

p9 l231 What is the three equation model for. Explain better.

p9 l233 Most of the Greenland glaciers don't have ice shelves at present. How does this scheme translate to the most common case of an outlet glacier ending in a vertical calving front? I think it doesn't. The models you discuss here have been developed for a typical Antarctic shelf geometry not for Greenland.

p9 l238 Remove "The first equation" and brackets around Eq 1 and similar for Eq 2 and 3 below.

p9-p10 I am confused about 3.2 as all of the discussed approaches are relevant for an Antarctic case but not for Greenland. Are we still in the Greenland use case, or is this about general model capability. If the latter, the paper needs to be restructured to discuss general modelling approaches aside from the concrete use case. See also general point on the question of paper scope.

p10 l258 Does "In our case" refer to the Greenland case? If so, the PICO model is not useful parameterisation for that case.

p10 l267 Why "not necessarily". I would say quite certainly not.

p10 l276 Start a new sentence between 'dynamics' and 'with'.

p10 l278 How would total ice sheet volume influence climate? Reformulate.

P11 l283 Where do the reconstructed ice sheet geometries originate from? Are these model states? Describe better.

P11 l286 Does the glacial cycle spinup use climate information from the same model? If not, any arguments about consistency? A glacial-interglacial ice sheet spin-up usually has the purpose to produce an internal ice rheology distribution in line with the history

of past forcing. Continuing with a steady forced ice sheet simulation destroys this information. How do you deal with this problem?

P11 l290 How do you examine the ice sheet volume, what is the criterium?

P11 l294 This asynchronous run was explained to violate physics. Modify that statement or explain why it is fine to do that here.

P11 l297 Translates to only 120 climate years, correct. Maybe add that as additional information.

P11 l302 Motivate your choice of 3 years. Why not more or less?

P11 l305 In my opinion it would be more interesting to see results of experiments 3 and 4 and not only in ice volume, but also e.g. in SMB components. Fig 4 seems to suggest that may outlet glaciers are thickening. Why is that?

P11 l306 It is good to see confirmed that the GrIS volume decreases with increasing boreal summer insolation, but also a pretty limited view of a complex coupled system. What else interesting is going on in these experiments? Are there any difference between different ice sheet sectors? What happens in with atmospheric circulation, the ocean and the sea-ice. How do outlet glaciers respond to those changes?

P12 l313 What are these assumptions and how are the results in line with those? Please also note my general point on the non-steady state behaviour of the Greenland ice sheet during the LIG.

P12 l316 "Figure 4"

P12 l316 I think it would make more sense and be more instructive to start the LIG and MH simulations from a fully coupled steady state PI and observe changes in all components as perturbations relative to that baseline.

P12 l323 Running an ice sheet model puts you in the position to identify the cause of a velocity change. Please confirm this statement from your model output.

p13 l369 Any discussion that could be added for figures 9 and 10? If not, I suggest to remove them from the manuscript.

p13 l372 What metrics is the model performance measured against?

p13 l371-376 Not clear what this comparison is supposed to show. I would believe the point is not to show that including a fully coupled ice sheet in a climate model has hardly any effect. I understand that it is good so see that including a dynamic Greenland ice sheet does not completely explode the climate. Nevertheless, there are other things worth exploring as I note here again: The standardised 4xCO2 experiment for example would be a great test to see if the ice sheet is retreating at a rate comparable to other models. Does the MOC respond at all to the additional freshwater input? How does the atmosphere see the changing ice sheet topography? Does the retreating ice sheet change the albedo? What are the feedbacks at play during ice sheet decay under strong atmospheric forcing? These are just the most basic questions that need to be addressed to convincingly show that the model is a useful tool for coupled simulations.

P14 l376 Is that seasonality specific for the model including the ice sheet, or is that a generic behaviour of the model? I would suggest to focus this section on aspects of the climate/ ice sheet system, that are different from the uncoupled climate model.

P14 l400 It seems strange that dEMB is mentioned for the first time in this manuscript in the conclusions.

p15 l412 Adapting coastlines seems like a long shot compared to all the other limitations of this model. Are there concrete ongoing works that address these issues?

Table 1. Why is this important? Is it discussed anywhere in the manuscript?

Figure 1. Explain better how topography (a) is an example of model resolution. If patches one sees are showing the resolution of the atmospheric grid, say so. This is not at all visible on a printout. Zooming in on the pdf until I see the patches, I see that they are at the image resolution limit. Suggest to enlarge and improve image

resolution.

Figure 2. Not clear what Runs 1 2 3 are. Are these years? Maybe indicate what SCOPE is in the upper figure, the green and blue arrows. It is not really clear where the separation is between the upper and the lower part.

Figure 3. 'Insolation anomalies for a the Mid Holocene and b the Last Interglacial compared to Pre-Industrial' Why not run MH and LIG from the PI ice sheet?

Figure 4. Why are many outlet glaciers thickening under LIG and MH climate? The PI ice sheet looks like filling the entire continent to the land-sea mask. That is typically the sign for an inadequate SMB boundary condition with way too little ablation. If this is the case, some critical statements are required here. The MH and LIG cases do not seem to show much if any retreat from the coast. This may be related to the point just before. No retreat during LIG. Red and black contours not visible on my printout nor on the pdf. Give units as colorbar labels. The ice Caption: The divide is not an area! The last sentence messes something.

Figure 5. The colour scale on the right panel suggests that there is no ablation area (all SMB is positive). Chose a better colour scale. The panels are too small on a printout. I don't see why data with global coverage has to lead to the odd rotation of the grid visible in the figure. Suggest to fix that for the sake of clarity of the figures.

Figure 6. Not sure what we should expect for the right hand panel given that the PICO model is clearly the wrong model for this purpose. But, if what we see is the only ocean forcing applied to this ice sheet, it is not very realistic, to say the least. I think at this point it is clear that this has to be discussed es a severe shortcoming of the model. With a more realistic representation of the interaction of Greenland outlet glaciers with the ocean, it would be interesting to see how far the ocean model grid extends, where ice is grounded and floating and what the extrapolated information is in between.

Figure 7. Can you give some explanation to what we see in this figure. What is the

origin of the inter-decadal variability visible in some periods? Are we looking at oscillations between two states? What is the reason for the arbitrary offset along the time axis?

Figure 8. It would be interesting to see more details about the effect of the coupling other than global temperature evolution. What is happening with the ice sheet in these runs and with the ocean and atmosphere around it?

Figure 9 and 10. Can be removed in my opinion, unless a meaningful discussion is added.

References

Fyke, J. G., Sacks, W. J., and Lipscomb, W. H.: A technique for generating consistent ice sheet initial conditions for coupled ice sheet/climate models, Geosci. Model Dev., 7, 1183–1195, https://doi.org/10.5194/gmd-7-1183-2014, 2014.

Fyke, J., Sergienko, O., Löfverström, M., Price, S., & Lenaerts, J. T. M. (2018). An overview of interactions and feedbacks between ice sheets and the Earth system. Reviews of Geophysics, 56, 361– 408. https://doi.org/10.1029/2018RG000600

Sellevold, R., van Kampenhout, L., Lenaerts, J. T. M., Noël, B., Lipscomb, W. H., and Vizcaino, M.: Surface mass balance downscaling through elevation classes in an Earth system model: application to the Greenland ice sheet, The Cryosphere, 13, 3193–3208, https://doi.org/10.5194/tc-13-3193-2019, 2019.

Smith, R. S., George, S., and Gregory, J. M.: FAMOUS version xotzb (FAMOUS-ice): a GCM capable of energy- and water- conserving coupling to an ice sheet model, Geosci. Model Dev. Discuss., https://doi.org/10.5194/gmd-2020-207, in review, 2020.

Vizcaíno, M., Mikolajewicz, U., Jungclaus, J., and Schurgers, G.: Climate modification by future ice sheet changes and consequences for ice sheet mass balance, Clim. Dynam., 34, 301–324, https://doi.org/10.1007/s00382-009-0591-y, 2010.

---

## Author Comment (AC1) · 30 Nov 2020

**Dear Dr. Huybrechts, dear Editorial Board, dear Reviewers,**

**We would like to thank you for taking the time to review our manuscript and provide useful and insightful comments which we plan to use in preparing a revised version for resubmission. Both reviewers had similar concerns, highlighting a lack of detail in describing the coupling system, as well as a too brief description of the climate states.**

**A difficulty we faced when putting this paper together was to separate the main methodological development from the actual case study experiments. That common, and admittedly justified, criticism central of both reviews is that this diffi-**

[Figure]

culty has not been overcome satisfactorily. Given that this conceptual problem also relates to many specific issues raised by the referees it seems to be best to split this manuscript into several parts: The part reviewed, discussed, and hopefully resubmitted here shall focus on the coupling methodology and any particular considerations applicable to AWI-ESM (in particular ECHAM6/JSBACH/FESOM1.4 and PISM 1.2.1), along with simple technical case studies. In upcoming studies, we will then examine specific details about the PMIP time slice simulations, allowing for a further, detailed discussion and an extended model-data comparison. We feel that this may significantly improve the separation between the various elements, and also give us a clear focus for each part of the study.

Enclosed below are specific responses to both the general comments as well as the specific comments raised by each of the reviewers. After a major revision, we would be delighted to again have the opportunity to submit our work, as we feel it is an important step in creating a modular, recyclable system for connecting various model comments.

Warm regards,

Dr. Paul Gierz

(on behalf of all co-authors)

**1 Review 1**

**1.1 Summary**

This paper describes aspects of the AWI-ESM-1.2 climate model, focussing specifically on the SCOPE system used to provide coupling to an interactive ice sheet component. It also gives some general information about the performance of the system in a set of case-studies. As the authors note, there is a pressing need to include ice sheet and shelf couplings to help answer many important questions in climate science, and this work addresses a timely and important topic. The SCOPE coupler at the heart of it all seems to be a flexible and useful tool likely to be of interest to workers in the field. Given the potential abilities of the model and coupler, and the range of simulations that have been conducted, I found overall that this manuscript provides a rather unsatisfying write-up which I think it could be significantly improved for readers by giving more detail in a number of areas.

We would like to thank you for your comments and taking the time to review our paper. Please find below some general responses which we shall consider upon submitting a revised manuscript. The minor comments, when obvious, will be fixed, and in case we feel there is a specific need to elaborate, we have included a response. We have marked in our responses which we plan on addressing in a revised version, and which may be saved for upcoming studies.

**1.2 General Comments**

Researchers in Earth System modelling are currently working out how best to implement ice sheet couplings in state-of-the-art models, and it's encouraging to see new models like this appearing. The SCOPE framework they describe seems like it is flexible enough to couple between various components, and also couple them at different

levels of complexity - eg SMB provided by either PDD-relevant fields or the dEBM - and I think that's a really useful approach that might enable other groups to try plugging an ice sheet into their climate modelling systems. I really like what it looks like they can do, and I'm pleased to see that the authors have included some paleo simulations as well as the DECK in the case studies they've done to illustrate its performance. However, I don't think this manuscript really gives me the level of detail I'd want to see in order to assess how AWI-ESM with ice sheets behaves in practice, nor contains enough to convince other modelling groups that they might want to use the SCOPE coupler. It's really valuable for interested readers to be able to see what the modelling systems do well, and also where they don't, as part of assessing what scientific questions its appropriate to use the model for. As it stands, the paper implies that the coupling system can pretty much do it all, but has only had those couplings tested in a very limited way, and the results of those tests are not evaluated here in much detail.

This is a short text, and pretty much the first third is given over to summarising preexisting model information. The coupling system is then described in terms that include the potential for a range of ice shelf-ocean interactions, and the Introduction has promised DECK and PMIP case-studies, so it's disappointing that the only details we get about how this all really plays out in practice concern Greenland-only runs in warm climates whose results are sketched out in the barest detail. As a GMD paper this isn't necessarily the place for a detailed description of simulation results for their own sake, but I think it's important that descriptions of a modelling technique come with enough illustration of how they work (and maybe how they don't) to be able to judge whether the proposed technique/model is useful for a particular application.

I may be misreading it, but the paper feels confused as to whether its purpose is to document how SCOPE and the coupling work in principle, or to describe how AWI-ESM behaves once equipped with ice sheets. Depending on the authors' goal, I might recommend adding different types of information to the paper. In either case, it wouldn't hurt to have a more focussed statement of the scope and aims of the work up front.

[Figure]

If SCOPE and the technical adjustments made to the models during coupling are the main focus - perhaps with a goal to advertise SCOPE to other groups - then I think they would improve the paper by providing more explicit sample configuration files that illustrate the flexibility in how it can be used and what transformations and parameterisations SCOPE can take care of and what would need to be altered inside the host atmosphere/land/ocean models. More illustrations of how SCOPE transforms input fields into information suitable for use in the target models for a range of case studies would be good here too. If instead the coupler performance really is inseparable from the AWI-ESM climate simulations it's embedded in, then it would be good to see a lot more /evaluation/ of what happens in those simulations, both to the ice and relevant climate fields - eg not just ice extents with respect to other work, but deep water formation in the ocean and how that affects large-scale heat transports. In an ideal world their case-studies would include some cold paleo climates and other ice sheets as well, to show a broader range of behaviour and sensitivity, but that is probably too much additional work to suggest at this point. Few details are shown in the case study simulations that are presented here, and there is almost no critical evaluation or comparison with other studies in what is presented, which is a little disappointing.

Indeed, it was a bit tricky to find the focus here. We want to both highlight the features of our new coupling methodology, but additionally also show first results with the system. However, the latter set of results should be viewed rather as feasibility tests: we want to show that our simulation system works, in principle. Of course, additional fine tuning and adjustment will be needed to fully understand the runs and ensure they are scientifically plausible, yet we would reserve those experiments and analyses for further, follow-up studies.

In this case, we will rewrite a large part of the methodology section and provide additional information regarding the SCOPE system, present examples of the coupled fields before and after the transformation, describe which transformations and corrections can be directly performed in SCOPE, and provide additional examples of the

SCOPE configuration files. We will strive to emphasize that the presented results of coupled climate / ice sheet rather represent feasibility studies than a full evaluation of the coupled climate / ice sheet system at various times of the geologic history, which would be beyond the framework of this manuscript that shall rather describe and introduce our new model system.

As mentioned above, the manuscript will then be split into several parts to better separate the methodology, which is the main development here, from the examination of the PMIP time slices.

**1.3 Detailed comments**

abstract: most of the abstract is concerned with motivating the inclusion of coupled ice sheets in climate models, rather than summarising the work that has been done here.

The abstract will be rearranged to include a specific focus on the new work done here, with the motivation left primarily in the introduction. (this paper)

line 5 (and elsewhere): the term "comprehensive" is used in a few places in an absolute sense, but no climate model includes representations of every possible - or even every relevant - physical process. This and "fully coupled" are simply inaccurate terms, unless used relatively to say that a model includes more couplings than others.

This will be changed to provide a better relation. In our context, we refer to Atmosphere-Biosphere-Ocean models as the "standard" General Circulation Models, while our new coupled model represents a "more comprehensive" Atmosphere-Biosphere-Ocean-Dynamic Inland Ice General Circulation Model. (this paper)

line 6 (and elsewhere): "cryosphere" is used in a few places in a way that makes most sense if taken to mean land-based ice, but the term is wider than that - snow cover and sea-ice are part of the cryosphere, for example, and their interactions are common features of most climate models

[Figure]

Will be changed to land-based ice throughout the revised manuscript. (this paper)

line 30: repeat of "large scale"

Will be removed.

line 40, 618: citation of Shi et al has year 0

BibTeX hiccup, thanks for letting us know.

line 53: having motivated the need for global ice sheet modelling, this is the first suggestion, almost off-hand, that only Greenland will be looked at in this work. This limitation is only actually made explicit on line 145 at the end of section 2.2 - I think it needs to be said much more clearly, much earlier, in the descriptions of what has been done in the abstract and introduction. Given that the system is apparently technically capable of looking at other ice sheets too, at least that's how I interpret line 146, I think that looking at only Greenland is a disappointingly limited case study. If more ambitious runs had been undertaken I guess that they would have been used instead, but it would have been great to see.

Here, we focused specifically on Greenland as a first use case. First tests for other domains (the entire Northern Hemisphere, Antarctica, and a dual-hemispheric setup) are under preparation, but we would like to reserve these experiments for follow up studies as including all these would be too broad for the focus of this paper. Preliminary examples could be included to demonstrate the flexibility of the system. We would include a Last Glacial Maximum setup, we would also show the Northern Hemisphere model domain. These would be presented preliminary in this paper, and in expanded detail in upcoming study, which would include an LGM simulation.

Sea level rise is of paramount interest for coastal communities around the globe. Therefore, the coupling of the Greenland ice sheet into coupled climate models offers the opportunity to address this question while Greenland's sea-level contribution is accelerating. Since the total mass balance of the Greenland Ice Sheet (GrIS) is predominately controlled by processes at the atmosphere-ice sheet interface, its coupling into AWI-CM provides an excellent showcase of how our coupling infrastructure allows seamless integration in climate models. The Antarctic case is of much higher complexity for several reasons which are beyond the SCOPE implementation. For instance, the still common unrealistic representation of the water mass modification by open-ocean convection leads to a significant temperature bias in the ocean temperatures adjacent to ice shelf caverns surrounding Antarctica. This bias would lead to unrealistic high basal melting rates within ice shelve caverns that drastically change the configuration of the Antarctic ice sheet. The presentation of the results and the analysis of the cause end effect for this specific point alone would be far beyond the current manuscript.

In addition, we know that the implementation of the precipitation boundary condition that determines Antarctica's ice gain is much more challenging than commonly anticipated: (Rodehacke, C. B., Pfeiffer, M., Semmler, T., Gurses, Ö., and Kleiner, T.: Precipitation Ansatz dependent Future Sea Level Contribution by Antarctica based on CMIP5 Model Forcing, Earth Syst. Dynam. Discuss., https://doi.org/10.5194/esd-2019-78, in review, 2020.; accepted). This uncertainty is important because climate models are in general subject to a positive precipitation bias (Palerme, Cyril, Christophe Genthon, Chantal Claud, Jennifer E. Kay, Norman B. Wood, and Tristan L'Ecuyer. 2017. "Evaluation of Current and Projected Antarctic Precipitation in CMIP5 Models." Climate Dynamics 48 (1–2): 225–39. https://doi.org/10.1007/s00382-016-3071-1.).

Furthermore, a comprehensive description of the ice discharge at the ice shelf margin is elusive. All these known unknowns besides others (earth viscosity differences between the East and West Antarctic Ice Sheet, uncertainties in geothermal heat flux distribution) constitute the unique challenges related to a coupling of Antarctica into climate models. Since these difficulties are outside of SCOPE infrastructure, we do not present any results where our climate model is technically coupled with an ice sheet model.

line 58: by "AR4 IPCC scenarios" do they mean the SRES scenarios (Nakicenovic
2000) - that might be a more general and helpful term here?

This will be changed for clarity. (upcoming study)

line 130: Gregory et al 2012 found that the size of artificial variability added to PDD melt calculations can have a major impact on the results - can more detail be given here than simply "white"? Gregory et al, "Modelling large-scale ice-sheet–climate interactions following glacial inception", Clim. Past, 8, 1565–1580, 2012

This is specific to how PISM implements the default PDD scheme, and we will clarify the difference between what SCOPE delivers, and how those actual fields are treated by the receiving model. (this paper)

line 161: "hinting" is a rather informal term. Why does the origin of the data have to be anonymised completely, it seems that having it potenitally retained in some form of metadata might be useful in some cases if SCOPE can be used to generate boundary condition files from different source models that could in principle be archived for later use?

The purpose of anonymising the actual data sent and received through SCOPE is to preserve modularity. In this case, any information that, for example ECHAM6, sends through SCOPE will simply appear as "atmosphere forcing". This gives us a 2-tiered structure in our coupling, and allows us to recycle elements of the interface. If, for example, another atmosphere model is selected for coupling to the ice sheet model via scope, only the configuration into SCOPE must be implemented. The elements received by PISM would then stay identical. However, it is a nice suggestion to include metadata regarding the origins of the fields, even if they are not actively used by, and part of, the system. (this paper)

line 186: I don't disagree with the conclusion that the ice sheets are likely not a major additional computational expense in the context of current CMIP ESMs, but it perhaps should be noted that the runs here only include Greenland at 5km. I imagine an Antarctic sheet/shelf system would cost significantly more, as would glaciated areas appropriate to a Last Glacial Maximum scenario, which I imagine is in scope since PMIP is an area of interest here. That thought raises another question - how would their system run separate GrIS and Antarctic ice sheets? In one global domain, or totally separate instances of PISM? Would that require separate SCOPE instances too? The timesteps and coupling intervals being used in the system have also not been cited, which might be useful.

From it's design, SCOPE can couple an arbitrary number of models. While we only demonstrate a single instance of a climate and an ice sheet model here, it is possible to send the same atmospheric information to multiple sources, e.g. one PISM domain for GrIS and another one for Antarctica. We will include an example configuration file showing how this could work. (this paper)

line 188: "violate the laws of physics" sounds extreme. I think outlining specific conservation issues implied by the asynchroneity would be better.

Indeed – we will revise this to specify the implications of an asynchronous approach. (this paper)

section 3.1: I'm not really clear on a couple of issues related to melt and conservation that run through this section. Perhaps they could be explained more clearly. I appreciate this stuff is tricky to explain - my description of my confusion that follows is not the clearest either, but should gave a flavour of the aspects I think could be rephrased.

As ECHAM/JSBACH runs, it must make some estimate of the melt over the ice sheet, if only as part of the latent heat calculation and estimating surface temperature. Is melt actually produced, and routed to the ocean, or is there no explicit melt/runoff occurring at all under an assumption (line 220) that the ice sheet (without interactivity) is in steady state so ice sheet runoff in JSBACH is just whatever precipitation has fallen?

This paragraph needs to be reworked to improve clarity, and will be generously rewritten in our next submission. To answer your questions: In the coupled setup, ice sheet mass loss, which is in our case treated as the sum of basal melting, surface melting, and calving, is added as **additional** water to the runoff scheme in JSBACH and transported, together with water volume originating from excess precipitation (that is also present in a model setup without a coupled ice sheet), to the ocean. In the uncoupled setup (e.g. without interactive ice sheets), the system is assumed to be in steady state: any precipitation falling on the ice sheet is treated first as snow, which then melts and directly flows to the ocean. There is no ice mass or loss from the actual ice sheet considered: e.g. in JSBACH, ice sheets are simply orographic features with a specific albedo and surface elevation (the latter is just implemented via the model's orography data set: this paper, with a section devoted to particular ECHAM/JSBACH - PISM coupling)

Under a PDD scheme, PISM will be calculating melt in a wholly different manner, un-connected with what happened in JSBACH - so that must be why the PISM ablation (line 198 - but under a PDD SMB scheme how is surface ablation different from runoff, is there basal water too?) is handed back and (line 224) the "hydrological discharge is corrected". That year's discharge has already happened in JSBACH though, so is an adjustment made to the amount of the next year's precipitation that makes its way to the ocean? Is that adjustment spread through the year evenly, or with some seasonal pattern?

In this case, there is no direct seasonality of the meltwater signal. We simply sum up the entire PISM run's mass loss (or gain), and average that over the entire period, then either increase (or decrease) the runoff sent to the ocean via JSBACH. Due to the nature of the coupling scheme (we serially alternate between the two models), this mass loss or gain is out of phase by up to the duration of one simulation run. In our feasibility experiments, we ran the climate model for 3 years, then the ice sheet model for 3 years; ergo the phase difference is up to 3 years. (this paper)

Does it have a spatial pattern? Is any attempt made to reconcile the surface latent heat

fluxes with the melting that PISM has seen?

The spatial pattern of the mass loss is given by PISM itself. In our case, if a particular edge of the GrIS melts more, that area will also see an additional increase in runoff relative to other areas. Surface latent heat fluxes are currently not corrected. (this paper: Specific considerations for ECHAM6/PISM)

line 205 says the orography changes are phased in over the next year of ECHAM run. Is the same true of the extent/mask changes, or are they instantaneous? In some models, the surface roughness (linked to the glaciated mask) would be related to the subgrid orography terms.

In this case, this is due to the nature of how ECHAM works: instantaneously changing the orography leads to numerical instability of the atmosphere model. If instead we phase in both geopotential height changes as well as subgrid-scale orography parameters over time, the model behaves in a stable manner. In contrast, the glacial mask is updated immediately. Consequently, the effect of ice sheets on radiation is present in AWI-ESM as soon as a previous coupling step from ice sheet to climate model has concluded, while the effect on atmospheric flow is slowly and stepwise imprinted during the year. We will update the manuscript text to outline these details. (this paper: Specific considerations for ECHAM6/PISM)

Linking to my question above, are the various water fluxes (line 212) phased through the year of climate run too? The discharge correction is not noted here, but it does say that surface mass loss is part of what is transported to the ocean - is this term actually the (corrected) ECHAM precipitation, or really the PISM surface ablation?

The actual term is the corrected ECHAM runoff (that is corrected for PISM mass loss or gain. (this paper)

Line 206 simply notes that the JSBACH glacial mask is updated. The PISM horizontal resolution is much finer than that in JSBACH - is there a threshold for the amount of

a JSBACH box that is glaciated in PISM used for this, or can JSBACH cope with both fractional ice extent in surface type and soil representations? Soil water is buried by advancing ice, but vegetation is destroyed - does JSBACH track carbon or nutrient in the soil, and is this buried as well? Is the vegetation mass/carbon simply lost from the model completely as ice advances, or conserved somehow?

In this case, we perform a bi-linear remapping onto the JSBACH grid, and cut off any cells which are less than 50% covered covered by ice, and additionally remove any areas thinner than 5 meters of ice thickness. The soil water is buried, and can be reintroduced in the case of ice retreat. In our version of the model, JSBACH does simulate land carbon, yet this is not actively coupled to the atmospheric carbon dioxide concentrations. Hence, what happens to carbon pools when ice sheets advance and retreat has no impact on the simulation results. Vegetation is simply removed when the ice sheet advances, and re-initialized with the potential to grow tundra type grasses occurs when the ice sheet retreats. These potential vegetation types are then changed by JSBACH, which determines the types of vegetation that can grow on grid cells based upon temperature, soil moisture, and precipitation. (this paper)

Line 215: calved mass would ideally not only be released over time, but over quite a large area as well - is all of this water placed at the surface, right at the coast?

In the current version of the model, that is the case. We are currently implementing a version which will have actual icebergs, thus improving upon the current limitation of the model setup. (this paper)

section 3.2: given that the previous section is so specific about what is done in the ECHAM-JSBACH case, it feel incongruous that this section talks mostly about capabilities that aren't used in AWI-ESM. It also only deals with the provision of a melt rate to the ice shelf, with nothing on information that might be passed back to the ocean, such as shelf geometry or adjustments for heat or salinity conservation based on meltrates that might have been computed in PISM or PICO. Can this be done in their system -

the land-surface equivalent is described in the previous section for ECHAM-JSBACH?

The focus of our paper is the coupling between the Greenland Ice Sheet and the climate model AWI-ESM. For this configuration, the ocean-ice sheet interaction is not the most leading process controlling Greenland's total ice mass. To our understanding, it is based on the fact that ice sheet surface processes dominate the recent ice loss. It is confirmed by existing estimates of Greenland's total mass balance (e.g., Sasgen et al., 2020, 2012). In addition, the bowl-shaped bedrock topography (Morlighem, 2017) supports stability against catastrophic ocean-driven collapses. Initially, we had planned to have the ocean model consider modifications of the ocean geometry. Still, this endeavor has turned out to be more challenging than originally anticipated. However, the coupling infrastructure is ready for these additional processes once their technical implementation has been finished.

Morlighem, M., C. N. Williams, E. Rignot, L. An, J. E. Arndt, J. L. Bamber, G. Catania, et al. 2017. "BedMachine v3: Complete Bed Topography and Ocean Bathymetry Mapping of Greenland From Multibeam Echo Sounding Combined With Mass Conservation." Geophysical Research Letters 44 (21): 11,051-11,061. https://doi.org/10.1002/2017GL074954.

Sasgen, Ingo, Bert Wouters, Alex S Gardner, Michalea D King, Marco Tedesco, Felix W Landerer, Christoph Dahle, Himanshu Save, and Xavier Fettweis. 2020. "Return to Rapid Ice Loss in Greenland and Record Loss in 2019 Detected by the GRACE-FO Satellites." Communications Earth & Environment 1 (1): 8. https://doi.org/10.1038/s43247-020-0010-1.

Shepherd, A., E.R. Ivins, A. Geruo, V.R. Barletta, M.J. Bentley, S. Bettadpur, K.H. Briggs, et al. 2012. "A Reconciled Estimate of Ice-Sheet Mass Balance." Science 338 (6111): 1183–89. https://doi.org/10.1126/science.1228102.

line 230: the first case described seems to be relevant for where the ocean model explicitly resolves shelf profile. Does SCOPE take 2d fields along the shelf surface, or

a 3d fields from the basin for this? When regridding, is any adjustment made for the greater resolution of surface geometry detail on the PISM grid, equivalent to the lapse rate adjustment to the surface temperatures in the PDD melt calculation?

We have an option where we can compute the melting rates by an independent model representing the interaction between ice and ocean. For instance, in this model, the basal melting is calculated at each depth level and interpolate vertically between the layers enclosing the depth of the actual interface between ocean and ice shelf base. Since the code exploits the linearized version of the ocean state equation (Equation 1 in Holland and Jenkins (1999)), the computation of the pressure-dependent melting temperature is not affected by this choice. The use of this option would allow us to consider differences in the basal ice shelf depth. However, for Greenland, subglacial water routing might be more critical – but unfortunately, this process is hardly understood. For example, the melt water released at the glacial faces, that can amplify the melting rates of fjord-terminating glaciers via turbulent processes by an order of magnitude, is probably more decisive than a small temperature shift due to height differences of the basal ice interface.

line 259: I'm not clear on where PICO sits in the framework here. Is it somehow built into SCOPE, or PISM? Does SCOPE prepare FESOM output for PICO, then get called again to process PICO output for PISM? If so, how are the separate parts called and coordinated - is coordination of sub-models a SCOPE function too?

PICO, which stands for *Potsdam Ice-shelf Cavity mOdel*, is part of the Parallel Ice Sheet Model PISM. We have clarified this point by stating: "In our case, we use SCOPE to prepare inputs for the Potsdam Ice-shelf Cavity mOdel (PICO; Reese et al., 2018), which is part of the Parallel Ice Sheet Model. PICO delivers the basal ice shelf conditions by exploiting 2-dimensional ocean temperature and salinity maps, which are, in our case, depth-averaged profiles."

line 267: further to the note on ocean vs ice sheet model resolution above, surely it's

not just that FESOM isn't resolving areas beyond the grounding line, but for the Greenland case-study used for the rest of the paper, it's not resolving /anything/ of the fjord systems that lead up to the marine-ice interfaces that PISM wants boundary conditions for. This is where PICO comes in as not just an option but absolutely essential.

We agree that the thermal ocean-driven ablation process is an integral part of the processes controlling ice loss. However, other configurations are possible besides the PICO option, such as computing the melting rates at the ice-ocean interface outside of PISM (we have this option available by an independent model code), or by using another parameterization available in PISM. Since our flexible coupling interface is open to all these configuration choices, we wouldn't say that PICO is necessary (not because the processes represented by PICO were irrelevant, but because its vital role can replaced by other model codes). Nevertheless, we have clarified this point by adding: "Besides the here described PICO option, SCOPE allows us to prepare forcing data for other parameterizations available in PISM, compute the melting rates at the ocean-ice interface with an independent model, or transfer melting rates coming from an ocean model representing this interface."

line 273: as noted before, it's a bit disappointing that only warm PMIP climates have been looked at. Could more comment be made on whether AWI-ESM could/will be used to look at colder paleo ice as well? The PI is also a bit tricky in terms of being their most modern case-study, it's much easier to provide an evaluation of the model performance for the present day for which there are actual observations.

AWI-ESM will indeed also be used for colder paleo time periods, and we are working on a Last Glacial Maximum simulation. Preliminary results of this simulation may be included in the revised manuscript, but detailed study of these results, as with the other time periods, will be saved for future publications.

line 274: this sentence is a good description of what I think should be in this section, but what is actually written in section 4 doesn't live up to this ambition yet.

Will be addressed in revised version.

line 302: I don't think it's said anywhere why a three year coupling period is chosen. I may have missed this detail, but line 130 notes that monthly data is saved from the atmosphere model - does the coupling pass a 3 year timeseries of atmospheric forcing data to PISM, or does it construct a climatology by averaging over those three years somehow?

This will be addressed in revised version. In the current implementation, both passing a timeseries or an average is possible. 3 years were selected as it seemed to be an optimum trade-off between computational walltime. Any shorter, and the PISM model would have taken less time than the couling itself.

line 317: I think a "Fig" is missing from the number in brackets "(4)". This figure is useful, but it would help the reader to see a more detailed evaluation of the LIG, MH and PI GrIS states compared to other reconstructions / simulations

Will be addressed in revised version.

line 337: The FESOM resolution is still not sufficient to resolve the majority of the coastal system relevant to the ice marine boundary conditions for the outlet glaciers, so although the ice/ocean grid numbers match up better there's still an awful lot of real physics missing. This is inevitable in a global model, but I think it should be stated more clearly. I'm not clear on how exactly PICO is deployed here either - are there individual (and individually tuned) PICO boxes for each outlet glacier, or are the different areas aggregated somehow? How has PICO been tuned for use here? Again, some evaluation of how the melt rates achieved stack up against other evidence, or even simply what is observed in the present day, would be useful.

In principle, due to its flexible way of discretizing space, we could configure FESOM with a resolution sufficient to resolve the major fjord systems with a width in the order of 10 km in the horizontal plane. However, stability considerations (the Courant–

Friedrichs–Lewy (CFL) condition) would require an extremely small time step that would prevent us from running simulations long enough to benefit from an interactive coupling between the climate system and land-based ice. We clarified the motivation of the chosen separation into distinct ocean regions and justified the use of the utilized PICO model by stating:

"The model PICO determines oceanographic forcing for defined regions that group smaller ice shelves or a large ice shelf within predefined regions labeled as "basins." In our case, this allows topographic and features, such as ocean currents, to differentiate between oceanographic conditions along the continental shelf and coastline of Greenland into the flowing ocean regions (Straneo et al., 2012):

**Northeast** Both the outflow of the Arctic Ocean, recirculation of Atlantic Water (Mouginot et al., 2015), and the circulation of the Greenland-Island-Norwegian Sea influence the region of the East Greenland Current (Våge et al., 2018) south of the Frame Strait and north of the Denmark Strait.

**Southeast** South of the Denmark Strait, the continuation of the East Greenland Current is formed by the outflow from the Greenland-Island-Norwegian Sea and Irminger Current (Harden et al., 2014; Inall et al., 2014). The latter is part of the North Atlantic subpolar gyre and carries the influence of tropical water masses (Daniault et al., 2016).

**Southwest** The Labrador Current and the reflected East Greenland Current and Irminger Current flow northward (Rignot et al., 2012).

**Northwest** North of the Davis Strait in the Baffin Bay, the coast currents continue flowing northward while the Davis Strait blocks the exchange of deeper water masses below $1000\,\text{m}$ depth (Rignot et al., 2012).

**North** It is part of the Arctic Ocean (Dickson et al., 2007) and connected via the shallow

Nares Strait (cross-section depth 220 m, Münchow et al., 2006) with the Baffin Bay (Johnson et al., 2011).

All around Greenland, deeper water masses circulate into fjords that lead to fjord-terminating glaciers, while topographic features, such as submarine ridges, control the flow of these water masses (e.g., Schaffer et al., 2020; Jackson and Straneo, 2016; Gladish et al., 2015; Sutherland et al., 2014; Mortensen et al., 2013). Across each ocean region/ice sheet basin, the ocean temperature and salinity are averaged. These averages drive the ocean melting in a region/basin, which depends on local conditions, such as the ice shelf depth of basal ice determining the ocean freezing point.

PICO's basic assumption of an ocean flow along an icy surface may also be valid for fjord-terminating glaciers because the interaction between seawater and ice generates overhanging ice fronts (Benn et al., 2007), as observations in Alaska (Sutherland et al., 2019) and Greenland confirm (Fried et al., 2015). These geometries resemble tiny ice shelf caverns. Since PICO requires that melting occurs at the grounding line, the related processes may mirror a turbulent driven melting in the plume-downstream direction (Jenkins, 2011). Nevertheless, the dominant obstacle for a detailed description of the ocean-driven melting is the full description of the ocean circulation in fjords that have a width in the order of $\mathcal{O}(10\,\mathrm{km})$. Even if we could resolve the ocean circulation, it would require better bathymetry maps because ridges control the exchange of water between the open ocean and the fjords (e.g., Schaffer et al., 2020; Jackson and Straneo, 2016; Gladish et al., 2015; Sutherland et al., 2014; Mortensen et al., 2013). Besides resolving ocean processes, also the subglacial routing of meltwater, which may amplify the melting by a factor of 10 (Xu et al., 2012, 2013), is not adequately resolved in the ice sheet model either. Considering all these uncertainties and limitations, we take a heuristic approach and apply PICO while aspiring better methods in the future.

line 351: typo, "asyncronously"

Will be fixed

line 354: are these simulations accelerated with the same protocol as used in part 3 of the "Spin-up Procedure", eg 3 climate years -> 25 ice years?

No, these runs are performed with 3 years climate and 3 years ice.

section 4.5: At less than 1 page, this section is simply too short to usefully illustrate or evaluate the model performance across two climate change scenarios and two paleoclimate simulations. Given that a coupled Greenland ice sheet is the main feature of the model in this paper, it is also a little bizarre that the section focusses on global, large-scale climate fields, with an evaluation limited to the fact that the coupled Greenland doesn't make much difference to the simulations. As it stands, it's little more than a statement of the simple fact that these climate setups can be run with AWI-ESM, it doesn't tell me anything about the ice sheet, how the coupling is working in the model or anything about the climate system in reality.

These runs were designed precisely for this: to demonstrate that the model functions in a fully coupled mode for PMIP timeslices. In a revised version, we will examine these experiments in a bit more detail, yet still only as feasibility studies. A full analysis of the experiments will the subject of future work.

line 411: I don't think "prognostically" is used correctly here. Not being able to change the coastlines mid-run is an important limitation, common to many models with coupled ice sheets, and should be mentioned earlier when the coupling system is first described.

fig 2: this schematic only shows atm to ice coupling, and omits the ocean/PICO coupling processes?

This was simply designed as an example figure, but the flow of data is similar for other coupled elements, e.g. FESOM–SCOPE–PICO.

The limitation you mentioned is a very general problem for all coupled models. In the

ice sheet model, the coastline can migrate at any time during its run due to shifts in the grounding line. Therefore, any related flooding of land or penetration of ice shelves into the ocean is possible at all times. In our case, the change of the coastline in the climate model would have to be done before the climate model restarts. We envision that once the ice sheet model is subject to a dramatic change, expressed, for instance, by the integrated number of changed model grid points converted from land into ocean points, the ice sheet model stops and allows the climate model to adjust to the change. In this respect, our coupling infrastructure supports all necessary steps, as appropriate hooks for changing restart files already exist. However, a full implementation for such a procedure has to be done in the ice sheet model and is therefore not part of the coupling infrastructure as such. Ongoing work to build such a system is being done and will be the subject of future publications.

fig 4: figure has no units

Will be corrected in the next submission

fig 5,6: caption doesn't note the source (eg simulation, time-averaging etc) for the coupling fields. It might be good to show these as block-fill plots rather than smooth contours, that would illustrate the resolutions that the different models are working at - 5a and b could show the ECHAM grid, and 5c the PISM one, for example.

Thanks for this suggestion. This figure may be revised to show both the sending and receiving side to illustrate the difference in model resolution. We also will consider changing the ocean figure, as the GrIS simulation shown in Figure 6 has little ocean interaction.

fig 9,10: figure has no units, caption should note the reference climate for the anomalies (preindustrial)

This will be corrected in a revised version.

**2 Review 2**

General comments: The focus of the paper has not become clear to me and there are contradicting signals in title, abstract and main text. Some parts read like the paper should be a description paper of the SCOPE coupler. Other parts suggest that it is a case study on fully- coupled simulations of the Greenland ice sheet. At the same time, references to other ice sheets (NHISs, AIS) are present suggesting that the work could be considered as establishing a fully coupled system, ready to be used for any (paleo) configuration. I see considerable shortcomings for the two latter interpretations, which makes me lean to suggesting a specific focus as a SCOPE description paper. In any case, the paper should be considerably reworked to make it clear from title to abstract and introduction what the focus of the paper is.

The manuscripts lacks important references to earlier and similar work on including interactive ice sheets in Earth System Models. At the same time, it seems that the authors do not fully appreciate the complexities associated with such undertaking (see e.g. Fyke et al., 2018). Considerable efforts have been made and are ongoing e.g. to improve the representation of the SMB over ice sheets (e.g. Vizcaíno et al., 2010; Sellevold et al, 2019) and to produce consistent coupled initial states for climate-ice sheet simulations (e.g. Fyke et al., 2014). Shortcomings of the current modelling approach should be critically discussed in view of these and other existing studies (e.g. Smith et al. 2020).

We would like to thank you for taking the time to provide a helpful and critical review of our manuscript. It was indeed tricky to find a good balance here, as we both want to technically describe our coupling strategy, as well as showcase studies of how the system can be applied in both projection and paleoclimate simulations. We are very well aware of the complexities of connecting ice sheet and climate simulations, and attempted to find a balance between describing a general purpose solution provided by the SCOPE coupler for connecting two model systems with suitable interfaces, as

well as describing how exactly this results in a coupled system in AWI-ESM. The recent literature suggested will be taken under consideration in a re-write, where we will attempt to focus the manuscript more on the technical possibilities afforded by the SCOPE coupler, as well as the specific considerations needed for the particular integration of PISM and ECHAM6/JSBACH/FESOM. We would like to emphasize that the case studies presented for the paleoclimate scenarios are simply that: case studies to demonstrate that the system functions. A full analysis of the climate and the transient nature of those periods will be reserved for upcoming studies.

It is mentioned in P6 l147 that other domains are implemented. But why are they not analysed? I don't think they can be considered similar enough so that showing the model for a Greenland case only is sufficient. AIS and NHISs have considerably different characteristics. In particular the interaction with the ocean of these marine ice sheets is clearly a different case than what can be done with a predominantly land-based Greenland during warm periods. The PICO model e.g. has been specifically developed for the Antarctic case. It is unclear to me why it is tested in the Greenland context. The interaction of the ocean with Greenland outlet glaciers is clearly not adequately represented in this model setup, which is a severe shortcoming for this use case that should be discussed.

This was also a point addressed by Reviewer #1. We are considering to include an example for the Last Glacial Maximum where we can demonstrate both that our system works for other ice sheet domains.

We agree that PICO has not been developed for Greenland. Considering all known limitations, such as

1. the missing representation of the fjord circulation controlling the warm water flow towards and meltwater from away from fjord-terminating glaciers,

2. too coarse or even unknown bathymetry data in fjords systems,

3. missing adequate representation of subglacial routing of meltwater towards the glacial terminus,

4. adequate release of the routed freshwater by one single, few distributed sources, or along the entire glacier face, description of the buffer capability of snow/firn to either form ice lenses or liquid water aquifers that buffer the release of meltwater, which influence the subglacial routing.

Since these known limitations combined may alter the melting rates by one to two orders of magnitude, we decided to take a heuristic approach by applying the PICO method. For further details please inspect the added motivation to the manuscript and our reply to reviewer #1.

If the aim of the paper is to show that the coupling is functional beyond a purely technical nature, it is crucial to see some critical experiments that explore the model's capabilities. How does the SMB for the present day over Greenland compare to observations and other model results? Is the SMB anywhere close to adequate as boundary condition for an ice flow model? If not, how does that limit the predictive capability of the model as a tool to look into warm climates of the past and future? How could the SMB be further improved? How does the atmosphere respond to an ice sheet that is considerably lowered and for an ice sheet that is retreating over land? How does the ocean respond to freshwater input? All of these are questions that need to be addressed should the model be used for simulations of the PD, LIG and future. And many more questions arise if the model should be employed for colder periods.

Considerable work on improving our SMB scheme has already been published (Krebs-Kanzow et al., 2018, 2020), and was still in the process of being included in the coupled system at the time of drafting the first version of this manuscript. Indeed, the PDD approach used in PISM in these experiments is limiting. During initial development, we performed several stress tests, which we will include in the revised version. Specifically, we are examining increasing and decreasing the ice sheet elevation to ensure a proper

atmospheric circulation response, checked for appropriate albedo responses when the ice sheet changes extent, and tested for salinity changes during melting events.

It should be noted somewhere prominently that the GrIS cannot be expected to be in a steady state neither for the LIG nor for the MH. We can of course use snapshot climate simulations to study the effect of a climate perturbation of a certain pattern and magnitude, but it should be clear that we are not looking at a real (transient) climate experiment. Comparing the GrIS sea-level contribution from these experiments with e.g. LIG reconstructions of sea-level is therefore problematic and requires some additional comments.

Indeed, the transient nature of GrIS ice volume makes it an inapplicable use case for snapshot style experiments. This will be highlighted in the discussion in a revised version.

Title:

From the title, it is not clear what SCOPE 1.0 is. Maybe "A case study using the Standalone Coupler SCOPE 1.0" The term "multi-resolution" in the title is not picked up in the manuscript. Suggest to remove it or add substance concerning this feature in the manuscript. The title mentions "ice sheets", but the paper presents only results for one ice sheet (Greenland). Suggest to rephrase. This will be corrected in a revised version.

Abstract: Important elements from the manuscript and title should be present in the abstract. The coupler SCOPE is not mentioned in the abstract, while it is an important part of the manuscript.

This will be corrected in a revised version.

P1 l19 I think you mean "future ... studies" as in "upcoming". Try to avoid paring "future" and "paleoclimate" in this sentence.

This will be corrected in a revised version.

Comments: P2 l25 It seems counter-intuitive to measure the amount of freshwater stored in the ice sheets in SLE. Could give the percentage of the global freshwater supply instead. Also not clear why the glaciers are included in this point.

We're sorry for this misunderstanding. In the glaciology community, it is not uncommon to express the ice stored on land as a potential sea-level contribution. We have followed this community-typical framing without considering the potential for a misunderstanding. These values will also be expressed as a relative amount of global freshwater in a revised version.

P2 L28 While you discuss the AIS, GrIS and glaciers before, here is only information about recent changes of the GrIS. Not clear why. Should extend to the AIS.

A brief section outline recent changes in the Antarctic ice sheet will be included.

p2 l30 Avoid repetition of "large-scale"

Noted, this will be fixed

P2 l31 What is "interior ocean circulation"? Rephrase? P2 l36 What are these shifts? Heinrich events, or DO events? P2 l37 "trigger or response". Neither trigger nor response suggest the notion that you put forward earlier of a fully coupled system with feedbacks. This may be important to reformulate.

All these points will be taken into account via rephrasing and added explanations.

P2 l38 "Earth System models with the capability"

Reviewer suggestion will be taken into account in a revised version.

P2 L39 Add version number after AWI-ESM?

This will be fixed in a revised version.

P2 L40 Remove "that is"

Fixed in a revised version.

P2 L40 Reference year Shi et al missing.

Noted, this shall be fixed.

P2 L42 "which is implemented"

Reviewer suggestion will be taken into account in a revised version.

P2 l45 "The simulations are based on"

Reviewer suggestion will be taken into account in a revised version.

P2 l48 "time slice simulations". Need to clarify how that relates transient coupled simulations.

Here we mean the typical, snap-shot style experiments often performed in the framework of PMIP-style model comparison projects. A transient experiment would vary greenhouse gases and orbital parameters during the run.

P2 l50 "climate states". Why plural?

Both the Holocene and Last Interglacial are interglacial climates.

P2 l51 Ice sheets can always have runoff. Do you mean anomalous runoff.

Yes, here we mean that runoff introduced into the ocean would result in a different response relative to a model without interactive ice sheets.

P2 l53 There is no motivation given why the focus is now solely on the GrIS. Please motivate that choice and why that is a good test case or easier to handle than the AIS or NHISs.

As mentioned, we focused on the GrIS here simply as a test case. Of course, examining also AIS or LIS/FIS would be interesting as well, and will be looked at in future studies.

P3 l76 This is a difficult sentence, consider introducing an abbreviation for ECMWF

Interactive
comment

before and reformulate.

Unfortunately, this is the model's actual full name: ECHAM = **E**uropean **C**entre for Medium-Range Weather Forecasts' Model in **Ham**urg.

P3 l84 Consider introducing abbreviation ECMWF before. See also previous point.

OK

P4 l116 Please describe what the implications of using dynamic vegetation are.

Using dynamic vegetation allows the vegetation fractions and cover types – and therefore also, the albedo – to adapt to the actual climate state. This is in particular interesting for the colder climate states such as the last glacial maximum, but is also of relevance in interglacial climate states. In particular, there is evidence that the Sahara may have supported vegetation during the Holocene (African humid period, see e.g. Otto-Bliesner et al., 2017, where the implications are discussed directly in the experiment design).

P5 l126 The PDD method is often assumed to be ill-fitted for paleo applications with different orbital parameters. This should be discussed as a possible caveat.

Indeed this is true. We will discuss this and also clearly state that PDD was only used for a feasibility test, but that future applications will use the dEBM, which does not have this limitation.

P5 l128 The units of the PDD factors are 3 orders of magnitude wrong. Typically 8 mm d-1 K-1 for ice.

Thank you for spotting this, that error will be corrected.

P5 l130 "forcing" is repeated here. Not clear what "forcing adds white noise" means Reformulate.

Many PDD schemes, including the one implemented in PISM, can employ input data

on monthly resolution. In order to mimic submonthly variations, white noise is added on top of the seasonal signal.

P5 l132 Sea-proximal. For Greenland this concerns marine-terminating outlet-glaciers. Reformulate?

This will be clarified in a revised version. This paragraph simply deals with the specific considerations we took into account when coupling the ocean and ice sheet models.

P5 l133 Are the three parameterisations employed simultaneously? Clarify.

Yes, all three are taken into account. It is common to employ multiple calving criteria.

P5 l146 The past sentence of the paragraph contradicts the sentence just before.

Here, we simply mean that we have already begun work on other ice sheet domains. Yet, these are not shown in this manuscript, and are reserved for upcoming studies. This will be reformulated in a new draft.

P6 l155 "shown in Figure 2"

Thanks, this will be fixed.

P6 l162 Could you please explain why the information needs to be anonymized?

This will be updated in a new draft. Here, we want to preserve the modularity of the system. In particular, coupling is divided into two steps: sending and receiving. Ideally, the model receiving information (e.g. the ice sheet model) does not need to know any details about the sending (e.g. atmosphere) side. Therefore, rather than having any particular information identifying the sender as ECHAM6, we simply denote the data as coming from an atmosphere model. This allows us to recycle the receiver side of the system even if the atmosphere system is changed to another model.

p6 l173 YAML needs a reference.

OK, we will include https://yaml.org in a revised version.

p6 l182 This sentence does not read well and contains too much information.

This will be removed or placed in the appendix in a revised version.

p6 l185 While the time spent for the coupler takes a relatively small percentage in the entire model, it must be much slower than an online coupling procedure within the model. It would be interesting to read some discussion about that and an estimate how much the coupling time could be reduced if a more efficient procedure would be employed.

It is important to note that the wall-time of the ice sheet model is considerably less than that of the remainder of the climate model. Since the Parallel Ice Sheet Model computes the time step width dynamically, it can vary by orders of magnitude. For example, rare intense calving events lead to a steep gradient of the ice surface. This gradient is the principal driver of the ice movement. As a consequence, the velocity reaches highest values requiring a small time step. For an online coupling, this feature would require to provide most of the time unused computer resources to prevent an even strong unbalanced case, where thousands of idle CPUs wait for the ice sheet model to finish. Therefore, an online coupling of ice sheet and of the remainder of the system would lead to a considerable reduction in the efficiency of the computation, since the nodes running the ice sheet simulation would spend a majority of the time idling. Offline coupling provides further advantages as well, as the interface can also be used "after the fact" – a completed run can be fed through the interface to generate a forced ice sheet run (albeit without resolving the feedbacks in the climate system).

P7 This is difficult to read (too small font) and does not provide critical information. Improve and move to the appendix?

A revised version of the manuscript will include more examples of SCOPE configuration files. These will be better integrated into the entire text to demonstrate how the coupling described in several of the initial feasibility tests can be set up. The poor quality was likely due to typesetting of the review version of the manuscript

P8 L189 It is not clear to me why and it sounds drastic to claim that running asynchronously means a violation of the laws of physics. Reformulate? I seem to understand that you do run the model in this way during initialisation, so this should be mentioned here as a case where it makes sense to employ the model in this way.

The point here is that it is impossible to conserve mass when running the model in a asynchronous manner. We are able to conserve the mass flux rates, but not the actual amounts. This will be reformulated in a revised version.

P8 l195 I miss a paragraph 3.0 about Atmosphere/Ice sheet coupling. How to produce an adequate SMB as boundary condition for an ice flow model is not at all obvious. The large difference in resolution between the atmosphere model and the ice sheet has to be bridged somehow. The fact that you use the PDD scheme in the ice sheet model does not make it easy to separate this problem, but should be discussed nevertheless. How is the atmospheric information interpolated/ downscaled. Does the PDD calculate in anomaly mode or with the absolute temperature. How do you deal with the fact that atmospheric grid cells can contain a mix of ice sheet, ice-free land and ocean?

A description of the coupling will be included in a revised version.

P8 l197 Be precise and explicit about what information is passed as change/anomaly and what as absolute field. What is the reference elevation (in the climate model and in the ice sheet model) if changes are communicated? Is the extent really communicated as a change in extent?

This will be fixed in a revised version. Changes are computed between the beginning and end of the PISM run, remapped onto the atmosphere model, and added to the previous atmosphere model's elevation. Extent is passed as any new glaciated areas above a minimum thickness, and then regridded conservatively to the atmospheric model grid.

P8 l198 Since you distinguish ablation and runoff, which field is used for what? WHat

do you use ablation for?

This is perhaps badly formulated. The ice sheet model distinguishes various mass loss terms (ablation, runoff, calving, basal melting), all of which are, in the current implementation, summed up and given as input to the hydrology scheme in the climate model.

P8 l200 Why is runoff routed by the atmosphere and not by the land surface model as mentioned on p3? Is this discussion not better placed in the interaction with the ocean? How do you distinguish runoff from subglacial discharge and from frontal/sub-shelf melt.

There may be a misunderstanding here. JSBACH (or, more precisely, the hydrol-ogy model, that is part of JSBACH) is embedded within ECHAM6: the two models ECHAM6 and JSBACH are not separable. When we speak of ECHAM6 we always mean ECHAM6/JSBACH as one component. This distinction will be unified in a re-vised version.

P8 l200 What is the hydrology scheme in AWI-ESM-1-1? Explain.

The hydrology scheme in AWI-ESM is embedded within JSBACH. The model has been developed by Hagemann et al. (1997). Here, net runoff is calculated and transported along pre-defined river flow directions to the coastal cells of the model. In the case of a coupled ice sheet model, mass loss from the ice sheet is included in this runoff scheme.

p8 l216 How does this removal from the hydrological scheme work. Is it globally dis-tributed?

No, this is local: Any mass gain in the ice domain is locally removed from the hydrolog-ical scheme, thus resulting in net less discharge.

p9 l230 What are the exchanged quantities? p9 l231 What is the three equation model for. Explain better.

The three equation model is one possible solution to determine ice shelf melting based upon temperature and salt properties. We invite the reviewer to examine Holland and Jenkins (1999) and Hellmer et al. (2003) for further details.

p9 l233 Most of the Greenland glaciers don't have ice shelves at present. How does this scheme translate to the most common case of an outlet glacier ending in a vertical calving front? I think it doesn't. The models you discuss here have been developed for a typical Antarctic shelf geometry not for Greenland.

As pointed out, the demonstrated ocean coupling here is designed simply to show the technical possibility of ice-shelf/ocean interaction. Indeed, the PICO model is not realistically applicable for Greenland. However, the the 3-eq model by Holland and Jenkins (199) is a very generic model. It describes the melting between an icey surface and ocean that is entirely driven by thermal forcing while it considers the impact of the pressure and salinity dependent freezing point temperature. Therefore, it is also applicable to any ice surface around Greenland under the assumption that melting is driven by a temperature difference between ice and ocean.

p9 l238 Remove "The first equation" and brackets around Eq 1 and similar for Eq 2 and 3 below.

This will be fixed in a revised version.

p9-p10 I am confused about 3.2 as all of the discussed approaches are relevant for an Antarctic case but not for Greenland. Are we still in the Greenland use case, or is this about general model capability. If the latter, the paper needs to be restructured to discuss general modelling approaches aside from the concrete use case. See also general point on the question of paper scope.

The next iteration will be reformulated to more clearly separate between the coupled system's (optional) capabilities, and what is actually used for a feasibility study

p10 l258 Does "In our case" refer to the Greenland case? If so, the PICO model is not

useful parameterisation for that case.

Same as above: this will be better sorted in a revised version.

p10 l267 Why "not necessarily". I would say quite certainly not.

This should in fact read "is unable to resolve sub-ice shelf caverns" rather than "not necessarily". FESOM does not currently have the option to explicitly simulate ice shelf caverns.

p10 l276 Start a new sentence between 'dynamics' and 'with'.

This will be fixed in a revised version.

p10 l278 How would total ice sheet volume influence climate? Reformulate.

This will be fixed in a revised version.

P11 l283 Where do the reconstructed ice sheet geometries originate from? Are these model states? Describe better.

This will be fixed in a revised version.

P11 l286 Does the glacial cycle spinup use climate information from the same model? If not, any arguments about consistency? A glacial-interglacial ice sheet spin-up usually has the purpose to produce an internal ice rheology distribution in line with the history of past forcing. Continuing with a steady forced ice sheet simulation destroys this information. How do you deal with this problem

The glacial spinup uses climate states for the LGM using ICE-6G reconstructions as well as for the Last Interglacial using a present day Greenland orography. This shall be clarified in a revised manuscript.

P11 l290 How do you examine the ice sheet volume, what is the criterium?

We wish to judge if the ice sheets are in steady state, thus no strong fluctuations in the ice sheet's total volume should be observed. We are aware that in the real climate system, ice sheets are likely never in equilibrium, yet to perform standard snapshot-style experiments, some approach must be adopted. We viewed this as being a feasible first step, and will discuss additional strategies and limitations

P11 l294 This asynchronous run was explained to violate physics. Modify that statement or explain why it is fine to do that here.

The spinup strategy will be better described and discussed (with benefits and pitfalls) in a revised version.

P11 l297 Translates to only 120 climate years, correct. Maybe add that as additional information.

Will be expanded upon in a revised draft.

P11 l302 Motivate your choice of 3 years. Why not more or less?

This was deemed as most computationally efficient. 1-1 coupling would render more time in memory allocation than actual simulation, and any longer periods of climate/ice would bring the physics out of sync.

P11 l305 In my opinion it would be more interesting to see results of experiments 3 and 4 and not only in ice volume, but also e.g. in SMB components. Fig 4 seems to suggest that may outlet glaciers are thickening. Why is that?

A more detailed analysis of SMB will be provided in a revised version. Likely, this has to do with the chosen SMB scheme between atmosphere and ice sheet. The revised version shall also examine different schemes (e.g. the dEBM mentioned in the paper).

P11 l306 It is good to see confirmed that the GrIS volume decreases with increasing boreal summer insolation, but also a pretty limited view of a complex coupled system. What else interesting is going on in these experiments? Are there any difference between different ice sheet sectors? What happens in with atmospheric circulation, the ocean and the sea-ice. How do outlet glaciers respond to those changes?

As mentioned earlier, the results presented for the PMIP time slices are designed as feasibility studies; we simply wish to illustrate that the model technically works. An in-depth examination of the changes, while no doubt interesting, shall be the subject of further papers, where we will have more room to expand and discuss the details.

P12 l313 What are these assumptions and how are the results in line with those? Please also note my general point on the non-steady state behaviour of the Greenland ice sheet during the LIG.

Shall be reworded. Here we meant that the sea level equivalent ice volume simulated by PISM are on the same order as those made sea levels that are published in the state of the art reconstructions.

P12 l316 "Figure 4"

Thanks, will be fixed.

P12 l316 I think it would make more sense and be more instructive to start the LIG and MH simulations from a fully coupled steady state PI and observe changes in all components as perturbations relative to that baseline.

This is difficult: even a fully coupled state for PI would necessitate simulating a full glacial history for the ice sheet, including feedbacks between the ice sheet and the remainder of the climate system. Given the throughput of our model, that is not feasible. As such, we elected to spin up the climate and ice sheet separately. A revised draft could look at the implications of this strategy.

P12 l323 Running an ice sheet model puts you in the position to identify the cause of a velocity change. Please confirm this statement from your model output.

This will be investigated in further detail in a revised draft.

p13 l369 Any discussion that could be added for figures 9 and 10? If not, I suggest to remove them from the manuscript.

Figures 9 and 10 will be reworked and better integrated in the revised manuscript.

p13 l372 What metrics is the model performance measured against?

Here, we examined specifically near-surface air temperatures as a global mean, to ensure that the basic climate state is in the same range as the uncoupled model.

p13 l371-376 Not clear what this comparison is supposed to show. I would believe the point is not to show that including a fully coupled ice sheet in a climate model has hardly any effect. I understand that it is good so see that including a dynamic Greenland ice sheet does not completely explode the climate. Nevertheless, there are other things worth exploring as I note here again: The standardised 4xCO2 experiment for example would be a great test to see if the ice sheet is retreating at a rate comparable to other models. Does the MOC respond at all to the additional freshwater input? How does the atmosphere see the changing ice sheet topography? Does the retreating ice sheet change the albedo? What are the feedbacks at play during ice sheet decay under strong atmospheric forcing? These are just the most basic questions that need to be addressed to convincingly show that the model is a useful tool for coupled simulations.

Thank you for suggesting to test the model in more extreme simulations. We will conduct a 4xCO2 simulation and run the model for a couple of hundred years to test the impact of such a strong carbon dioxide forcing on ice sheets and the impact of the resulting meltwater discharge on ocean circulation. The results will be included in the revised manuscript.

P14 l376 Is that seasonality specific for the model including the ice sheet, or is that a generic behaviour of the model? I would suggest to focus this section on aspects of the climate/ ice sheet system, that are different from the uncoupled climate model.

This case also holds for the uncoupled climate model. A revised version will address in particular differences between coupled and uncoupled experiments; yet as mentioned, these PMIP style runs are designed primarily as feasibility tests, rather than for serious

scientific evaluation, which will be the topic of future studies.

P14 l400 It seems strange that dEMB is mentioned for the first time in this manuscript in the conclusions.

We shall make sure that schemes employed in the model will be detailed more explicitly in the model description.

p15 l412 Adapting coastlines seems like a long shot compared to all the other limitations of this model. Are there concrete ongoing works that address these issues?

Adaptive land-sea masks will be of prime importance if real deglaciation runs are to be conducted. This is currently under active development.

Table 1. Why is this important? Is it discussed anywhere in the manuscript

As our model is able to simulate dynamic vegetation, which may change in response to different climates. This is a key for both albedo as well as providing one difficulty that needed to be overcome during coupling – which values shall be initialized when the ice sheet retreats? As discussed in the manuscript, we initialize newly deglaciated land points with tundra, and remove vegetation if the ice sheet advances.

Figure 1. Explain better how topography (a) is an example of model resolution. If patches one sees are showing the resolution of the atmospheric grid, say so. This is not at all visible on a printout. Zooming in on the pdf until I see the patches, I see that they are at the image resolution limit. Suggest to enlarge and improve image resolution.

Figure 1 is designed to show the horizontal grid resolution of each model. The "zoom" error may be due to GMD's quality requirements, which did not allow us to upload high resolution figures. We will contact the editorial department to correct this in the next draft.

Figure 2. Not clear what Runs 1 2 3 are. Are these years? Maybe indicate what

SCOPE is in the upper figure, the green and blue arrows. It is not really clear where the separation is between the upper and the lower part.

A large portion of the figure described here will be dealt with in a new version of the manuscript, where an entire section of the technical coupling will be presented. Here, Runs 1, 2, 3 are simply arbitrarily simulations, each with $n$ years, all equally long. The figure shall be improved upon.

Figure 3. 'Insolation anomalies for a the Mid Holocene and b the Last Interglacial compared to Pre-Industrial' Why not run MH and LIG from the PI ice sheet?

Both the MH and LIG simulations were initialized from the PI spinup state. To clarify, this will be rephrased in a new draft.

Figure 4. Why are many outlet glaciers thickening under LIG and MH climate? The PI ice sheet looks like filling the entire continent to the land-sea mask. That is typically the sign for an inadequate SMB boundary condition with way too little ablation. If this is the case, some critical statements are required here. The MH and LIG cases do not seem to show much if any retreat from the coast. This may be related to the point just before. No retreat during LIG. Red and black contours not visible on my printout nor on the pdf. Give units as colorbar labels. The ice Caption: The divide is not an area! The last sentence messes something.

The thickening of the ice sheet may be related, in part, to an inadequate SMB scheme, something we will delve more deeply into in the next iteration. Considerably work has been done on improving the SMB scheme via the dEBM model, which will also be showcased in a revised draft. Missing contours may be a "low-resolution" problem required by the submission process for GMD. Units shall be included in the colorbar labels. The last sentence should read "...elevations below 500 m are masked.

Figure 5. The colour scale on the right panel suggests that there is no ablation area (all SMB is positive). Chose a better colour scale. The panels are too small on a printout.

I don't see why data with global coverage has to lead to the odd rotation of the grid visible in the figure. Suggest to fix that for the sake of clarity of the figures.

This is a plotting error, and will be corrected in a revised version. Shown here were (accidentally) simply the values for December, rather than a yearly integral.

Figure 6. Not sure what we should expect for the right hand panel given that the PICO model is clearly the wrong model for this purpose. But, if what we see is the only ocean forcing applied to this ice sheet, it is not very realistic, to say the least. I think at this point it is clear that this has to be discussed es a severe shortcoming of the model. With a more realistic representation of the interaction of Greenland outlet glaciers with the ocean, it would be interesting to see how far the ocean model grid extends, where ice is grounded and floating and what the extrapolated information is in between.

This figure will be replaced in a newer version. Indeed, PICO is likely not applicable for the Northern Hemisphere, and this figure was merely included to demonstrate the technical workings of the coupling system. We shall focus instead on PISM ocean implementation which is applicable for the Northern hemisphere.

Figure 7. Can you give some explanation to what we see in this figure. What is the origin of the inter-decadal variability visible in some periods? Are we looking at oscillations between two states? What is the reason for the arbitrary offset along the time axis?

The offset in the time axis is a plotting error, and will be corrected in a revised version. The variability is a feature of the spinup procedure, where a forcing is constantly applied for a particular ice time period. The variability is likely due to the random noise applied in PISM's built-in PDD scheme.

Figure 8. It would be interesting to see more details about the effect of the coupling other than global temperature evolution. What is happening with the ice sheet in these runs and with the ocean and atmosphere around it?

These figures can be amended to also include global albedo and sea-ice area to gain a better insight into what the coupling is influencing.

Figure 9 and 10. Can be removed in my opinion, unless a meaningful discussion is added.

These figures will be replaced with more focused analyses of climate features directly connected to the interactive ice sheets.

**References**

Benn, D. I., Warren, C. R., and Mottram, R. H.: Calving processes and the dynamics of calving glaciers, Earth-Science Reviews, 82, 143–179, https://doi.org/10.1016/j.earscirev.2007.02.002, http://linkinghub.elsevier.com/retrieve/pii/S0012825207000396, 2007.

Daniault, N., Mercier, H., Lherminier, P., Sarafanov, A., Falina, A., Zunino, P., Pérez, F. F., Ríos, A. F., Ferron, B., Huck, T., Thierry, V., and Gladyshev, S.: The northern North Atlantic Ocean mean circulation in the early 21st century, Progress in Oceanography, 146, 142–158, https://doi.org/10.1016/j.pocean.2016.06.007, http://dx.doi.org/10.1016/j.pocean.2016.06.007http://linkinghub.elsevier.com/retrieve/pii/S0079661116300659, 2016.

Dickson, R., Rudels, B., Dye, S., Karcher, M., Meincke, J., and Yashayaev, I.: Current estimates of freshwater flux through Arctic and subarctic seas, Progress In Oceanography, 73, 210–230, https://doi.org/10.1016/j.pocean.2006.12.003, http://linkinghub.elsevier.com/retrieve/pii/S007966110700081X, 2007.

Fried, M. J., Catania, G. A., Bartholomaus, T. C., Duncan, D., Davis, M., Stearns, L. A., Nash, J., Shroyer, E., and Sutherland, D.: Distributed subglacial discharge drives significant submarine melt at a Greenland tidewater glacier, Geophysical Research Letters, 42, 9328–9336, https://doi.org/10.1002/2015GL065806, http://doi.wiley.com/10.1002/2015GL065806, 2015.

Gladish, C., Holland, D., Rosing-Asvid, A., Behrens, J., and Boje, J.: Oceanic Boundary Conditions for Jakobshavn Glacier. Part I: Variability and Renewal of Ilulissat Icefjord Waters, 2001–14, Journal of Physical Oceanography, 45, 3–32, https://doi.org/10.1175/JPO-D-14-0044.1, 2015.

Harden, B., Pickart, R., and Renfrew, I.: Offshore Transport of Dense Water from the East

Greenland Shelf, Journal of Physical Oceanography, 44, 229–245, https://doi.org/10.1175/JPO-D-12-0218.1, 2014.

Inall, M. E., Murray, T., Cottier, F. R., Scharrer, K., Boyd, T. J., Heywood, K. J., and Bevan, S. L.: Oceanic heat delivery via Kangerdlugssuaq Fjord to the south-east Greenland ice sheet, Journal of Geophysical Research: Oceans, 119, 15pp, https://doi.org/10.1002/2013JC009295, http://doi.wiley.com/10.1002/2013JC009295, 2014.

Jackson, R. H. and Straneo, F.: Heat, Salt, and Freshwater Budgets for a Glacial Fjord in Greenland, Journal of Physical Oceanography, 46, 2735–2768, https://doi.org/10.1175/JPO-D-15-0134.1, http://journals.ametsoc.org/doi/abs/10.1175/JPO-D-15-0134.1, 2016.

Jenkins, A.: Convection-driven melting near the grounding lines of ice shelves and tidewater glaciers, Journal of Physical Oceanography, 41, 2279–2294, https://doi.org/10.1175/JPO-D-11-03.1, http://journals.ametsoc.org/doi/abs/10.1175/JPO-D-11-03.1, 2011.

Johnson, H., Münchow, A., Falkner, K., and Melling, H.: Ocean circulation and properties in Petermann Fjord, Greenland, Journal of Geophysical Research, 116, 18pp, https://doi.org/10.1029/2010JC006519, http://www.agu.org/pubs/crossref/2011/2010JC006519.shtml, 2011.

Mortensen, J., Bendtsen, J., Motyka, R., Lennert, K., Truffer, M., Fahnestock, M., and Rysgaard, S.: On the seasonal freshwater stratification in the proximity of fast-flowing tidewater outlet glaciers in a sub-Arctic sill fjord, Journal of Geophysical Research, 118, 14pp, https://doi.org/10.1002/jgrc.20134, http://onlinelibrary.wiley.com/doi/10.1002/jgrc.20134/abstract, 2013.

Mouginot, J., Rignot, E., Scheuchl, B., Fenty, I., Khazendar, A., Morlighem, M., Buzzi, A., and Paden, J.: Fast retreat of Zachariae Isstrom, northeast Greenland, Science, p. 8pp, https://doi.org/10.1126/science.aac7111, http://www.sciencemag.org/cgi/doi/10.1126/science.aac7111, 2015.

Münchow, A., Melling, H., and Falkner, K. K.: An Observational Estimate of Volume and Freshwater Flux Leaving the Arctic Ocean through Nares Strait, Journal of Physical Oceanography, 36, 2025–2041, https://doi.org/10.1175/JPO2962.1, https://journals.ametsoc.org/jpo/article/36/11/2025/10519/An-Observational-Estimate-of-Volume-and-Freshwater, 2006.

Rignot, E., Fenty, I., Menemenlis, D., and Xu, Y.: Spreading of warm ocean waters around Greenland as a possible cause for glacier acceleration, Annals of Glaciology, 53, 257–266, https://doi.org/10.3189/2012AoG60A136, http://www.igsoc.org/annals/53/60/a60A136.html, 2012.

Schaffer, J., Kanzow, T., von Appen, W.-J., von Albedyll, L., Arndt, J. E., and Roberts, D. H.:

Bathymetry constrains ocean heat supply to Greenland's largest glacier tongue, Nature Geoscience, 1, 8pp, https://doi.org/10.1038/s41561-019-0529-x, http://www.nature.com/articles/s41561-019-0529-x, 2020.

Straneo, F., Sutherland, D. A., Holland, D., Gladish, C., Hamilton, G. S., Johnson, H. L., Rignot, E., Xu, Y., and Koppes, M.: Characteristics of ocean waters reaching Greenland's glaciers, Annals of Glaciology, 53, 202–210, https://doi.org/10.3189/2012AoG60A059, http://www.igsoc.org/annals/53/60/a60A059.html, 2012.

Sutherland, D. A., Straneo, F., and Pickart, R. S.: Characteristics and dynamics of the two major Greenland glacial fjords, Journal of Geophysical Research: Oceans, 119, 25pp, https://doi.org/10.1002/2013JC009786, http://doi.wiley.com/10.1002/2013JC009786, 2014.

Sutherland, D. A., Jackson, R. H., Kienholz, C., Amundson, J. M., Dryer, W. P., Duncan, D., Eidam, E. F., Motyka, R. J., and Nash, J. D.: Direct observations of submarine melt and subsurface geometry at a tidewater glacier, Science, 365, 369–374, https://doi.org/10.1126/science.aax3528, http://www.sciencemag.org/lookup/doi/10.1126/science.aax3528, 2019.

Våge, K., Papritz, L., Håvik, L., Spall, M. A., and Moore, G. W. K.: Ocean convection linked to the recent ice edge retreat along east Greenland, Nature Communications, 9, 1287, https://doi.org/10.1038/s41467-018-03468-6, http://www.nature.com/articles/s41467-018-03468-6, 2018.

Xu, Y., Rignot, E., Menemenlis, D., and Koppes, M.: Numerical experiments on subaqueous melting of Greenland tidewater glaciers in response to ocean warming and enhanced subglacial discharge, Annals of Glaciology, 53, 229–234, https://doi.org/10.3189/2012AoG60A139, http://www.igsoc.org/annals/53/60/t60A139.html, 2012.

Xu, Y., Rignot, E., Fenty, I., Menemenlis, D., and Mar Flexas, M.: Subaqueous melting of Store Glacier, West Greenland from three-dimensional, high-resolution numerical modeling and ocean observations, Geophysical Research Letters, 40, 6pp, https://doi.org/10.1002/grl.50825, http://doi.wiley.com/10.1002/grl.50825, 2013.